# Prohibitin 1 is essential to preserve mitochondria and myelin integrity in Schwann cells

Gustavo Della-Flora Nunes [1,2], Emma R. Wilson [1,2], Leandro N. Marziali[1,2], Edward Hurley[1], Nicholas Silvestri[3], Bin He [4], Bert W. O'Malley [5], Bogdan Beirowski [1,2], Yannick Poitelon[6], Lawrence Wrabetz[1,2,3] & M. Laura Feltri [1,2,3✉]

In peripheral nerves, Schwann cells form myelin and provide trophic support to axons. We previously showed that the mitochondrial protein prohibitin 2 can localize to the axon-Schwann-cell interface and is required for developmental myelination. Whether the homologous protein prohibitin 1 has a similar role, and whether prohibitins also play important roles in Schwann cell mitochondria is unknown. Here, we show that deletion of prohibitin 1 in Schwann cells minimally perturbs development, but later triggers a severe demyelinating peripheral neuropathy. Moreover, mitochondria are heavily affected by ablation of prohibitin 1 and demyelination occurs preferentially in cells with apparent mitochondrial loss. Furthermore, in response to mitochondrial damage, Schwann cells trigger the integrated stress response, but, contrary to what was previously suggested, this response is not detrimental in this context. These results identify a role for prohibitin 1 in myelin integrity and advance our understanding about the Schwann cell response to mitochondrial damage.

[1] Hunter James Kelly Research Institute, Jacobs School of Medicine and Biomedical Sciences, University at Buffalo, Buffalo, NY, USA. [2] Departments of Biochemistry, Jacobs School of Medicine and Biomedical Sciences, University at Buffalo, Buffalo, NY, USA. [3] Departments of Neurology, Jacobs School of Medicine and Biomedical Sciences, University at Buffalo, Buffalo, NY, USA. [4] Immunobiology & Transplant Science Center and Department of Surgery, Houston Methodist Hospital, Houston, TX, USA. [5] Departments of Medicine and Molecular and Cellular Biology, Baylor College of Medicine, Houston, TX, USA. [6] Albany Medical College, Dept of Neuroscience and Experimental Therapeutics, Albany, NY, USA. ✉email: mlfeltri@buffalo.edu

In the peripheral nervous system (PNS), Schwann cells (SCs) surround axons with myelin, a multi-layered lipid-rich membrane essential for rapid and efficient propagation of nerve signals. The myelination process is metabolically demanding, requiring massive biosynthesis of lipids and proteins in a short period of time. It is estimated that the average SC total membrane area (plasma membrane + myelin) expands 2600-fold from day 1 to day 75 of rat development[1]. Furthermore, axons of the PNS are either completely myelinated or fully surrounded by non-myelinating SCs, which severely limits direct contact between the axons and the extracellular environment. Therefore, SCs are also believed to provide axons with metabolic and trophic support, a function that seems to be independent of myelination[2]. As a consequence, SC metabolism likely needs to be tightly regulated to allow both myelination and axonal support.

Mitochondria, which are pivotal organelles in cellular metabolism, are quickly emerging as important regulators of myelin formation and maintenance in SCs. Peripheral neuropathy is a common manifestation of genetic mitochondrial disorders, affecting about 30% of all patients[3]. Moreover, in those cases where the neuropathy is demyelinating, morphological alterations frequently concentrate in mitochondria of SCs rather than axons, suggesting a link between mitochondrial dysfunction in SCs and demyelination[4]. Recently, two different studies showed that deletion of mitochondrial genes in SCs results in peripheral neuropathy in mice[5,6]. However, the mechanisms by which dysfunctional SC mitochondria lead to demyelination remain incompletely understood. Viader et al. suggested that activation of a maladaptive integrated stress response (ISR) is partially causal for demyelination downstream of mitochondrial dysfunction[7], but this hypothesis has not been thoroughly investigated.

We recently identified prohibitin 2 (PHB2) as an essential protein for developmental myelination in mice[8]. Moreover, we found evidence that PHB2 can localize to the axon-SC interface and is important in the early contact between these two cell types[8]. Although prohibitins have been reported to reside in diverse subcellular locations, they are most well known for their role in mitochondria, where they carry out most of their functions[9]. In the mitochondria, PHB2 associates with the homologous protein PHB1, forming a large oligomeric ring-like structure in the inner mitochondrial membrane[10]. This structure is thought to act as a platform that aids in the regulation of several aspects of mitochondrial biogenesis, dynamics, and metabolism[9,11]. Thus, we set out to investigate the role of PHB1 in SCs in vivo.

Here, we show that, contrary to deletion of Phb2, deletion of Phb1 in SCs causes only a minimal developmental phenotype, but triggers a severe demyelinating peripheral neuropathy after myelination is completed. A careful characterization of potential functions of PHB1 revealed that mitochondria were heavily affected and that mitochondrial damage was progressive and accumulated in single cells. Interestingly, we found a sudden loss of mitochondria in discrete cells, which was associated with demyelination. We also confirmed that the cellular response to mitochondrial damage in SCs indeed involves the ISR, but, contrary to what has been previously suggested, we demonstrate that the ISR is not detrimental in the context of demyelination induced by mitochondrial injury in SCs. These results advance our understanding of how SCs respond to mitochondrial damage, solidify the importance of SC mitochondria to maintain nerve homeostasis, and reveal that, unexpectedly, PHB1 and PHB2 may have some independent functions.

## Results

**SC-specific knockout of *Phb1*.** Given the involvement of PHB2 in developmental myelination[8], we sought to investigate the role that

PHB1 plays in SCs in vivo. To this end, we crossed mice bearing a floxed *Phb1* gene[12] with mice expressing *Cre* recombinase under the control of the *Mpz* promoter[13]. This allowed us to generate mice in which *Phb1* was deleted specifically in SCs (*Phb1*^fl/fl^; *Mpz-Cre*—referred to as Phb1-SCKO throughout) (Fig. 1a). Recombination in sciatic nerves of Phb1-SCKO animals was confirmed by PCR (Fig. 1b) and resulted in a significant reduction of *Phb1* mRNA (Fig. 1c) and protein (Supplementary Fig. 1a and 1b).

**Deletion of prohibitin 1 in SCs triggers a severe peripheral neuropathy.** We then examined the morphology of sciatic nerves of Phb1-SCKO mice at different days of post-natal development: post-natal day (P)10, P20, P40, P60, P90, and P120. Ablation of *Phb1* does not impair radial sorting, (the process by which larger caliber axons are selected to be myelinated) as occurs following the deletion of *Phb2* in SCs[8], but instead leads to delayed myelination with most SCs still at the pro-myelinating stage in P10 animals (Supplementary Fig. 1c and 1d). However, by P20, Phb1-SCKO mice have an equivalent number of myelinated axons when compared to controls (Fig. 1d, e), although they are slightly hypomyelinated (Supplementary Fig. 1e). Strikingly, this almost complete recovery from the developmental delay is then followed by rapid and profound demyelination, which leads to a 60% reduction in the number of myelinated fibers in sciatic nerves of Phb1-SCKO mice between P20 and P60 (Fig. 1d, e). Morphologically, demyelination is suggested by the presence of several large axons devoid of myelin and by SCs containing myelin debris in cytosolic compartments (Fig. 1d–f). Concomitant with the demyelination, we also identified signs of axonal degeneration, such as axonal shrinkage and local accumulation of vesicles and organelles, indicative of transport defects (Fig. 1f). In addition, Phb1-SCKO mice crossed to a Thy-1 YFP axonal reporter line present with axonal swellings and axon fragmentation in tibial nerves at P20 and P40, respectively (Fig. 1g). This peripheral neuropathy also leads to clear functional impairments in Phb1-SCKO mice. Mutant mice show reduced nerve conduction velocity at P20 and P40 and decreased compound muscle action potential (CMAP) amplitude at P40 (Fig. 1h), in addition to reduced motor performance in the rotarod test at P20 (Supplementary Fig. 1f). In line with the typical manifestation of peripheral neuropathy in mice, Phb1-SCKO animals show clenching of hind limbs towards the body when suspended by the tail (Supplementary Fig. 1g). Moreover, these animals show gait impairments that progress to hind limb paresis or paralysis (Supplementary Video 1).

**Expression of PHB1 and PHB2 is partially interdependent.** PHB1 and PHB2 are thought to carry out most of their functions together as heterodimers[10,14]. Therefore, it was very surprising that Phb1-SCKO and Phb2-SCKO mice showed significant phenotypic differences, with Phb2 seemingly being more essential for SC development. We first asked if this difference could be explained by a distinct expression pattern of these proteins. By mRNA, the expression of prohibitins was highest early in developing nerves (P1 and P5) and progressively decreased as the animals aged, with *Phb2* showing a steeper reduction compared to *Phb1* (Fig. 2a). This faster reduction in Phb2 levels could reflect a more important developmental role of Phb2 in SC. On the other hand, PHB1 and PHB2 protein levels seemed to be linked, peaking at around P5 and diminishing in older animals (Fig. 2b, c). Thus, the expression pattern of Phb1 and Phb2 shows a very similar trajectory over time. Next, we investigated whether the expression of *Phb2* is altered when *Phb1* is ablated in SCs. In P20 Phb1-SCKO animals, Phb2 mRNA levels do not change (Fig. 2d), in contrast with the reported decrease in Phb1 mRNA levels at the same age (Fig. 1c). On the other hand, PHB2 protein levels are

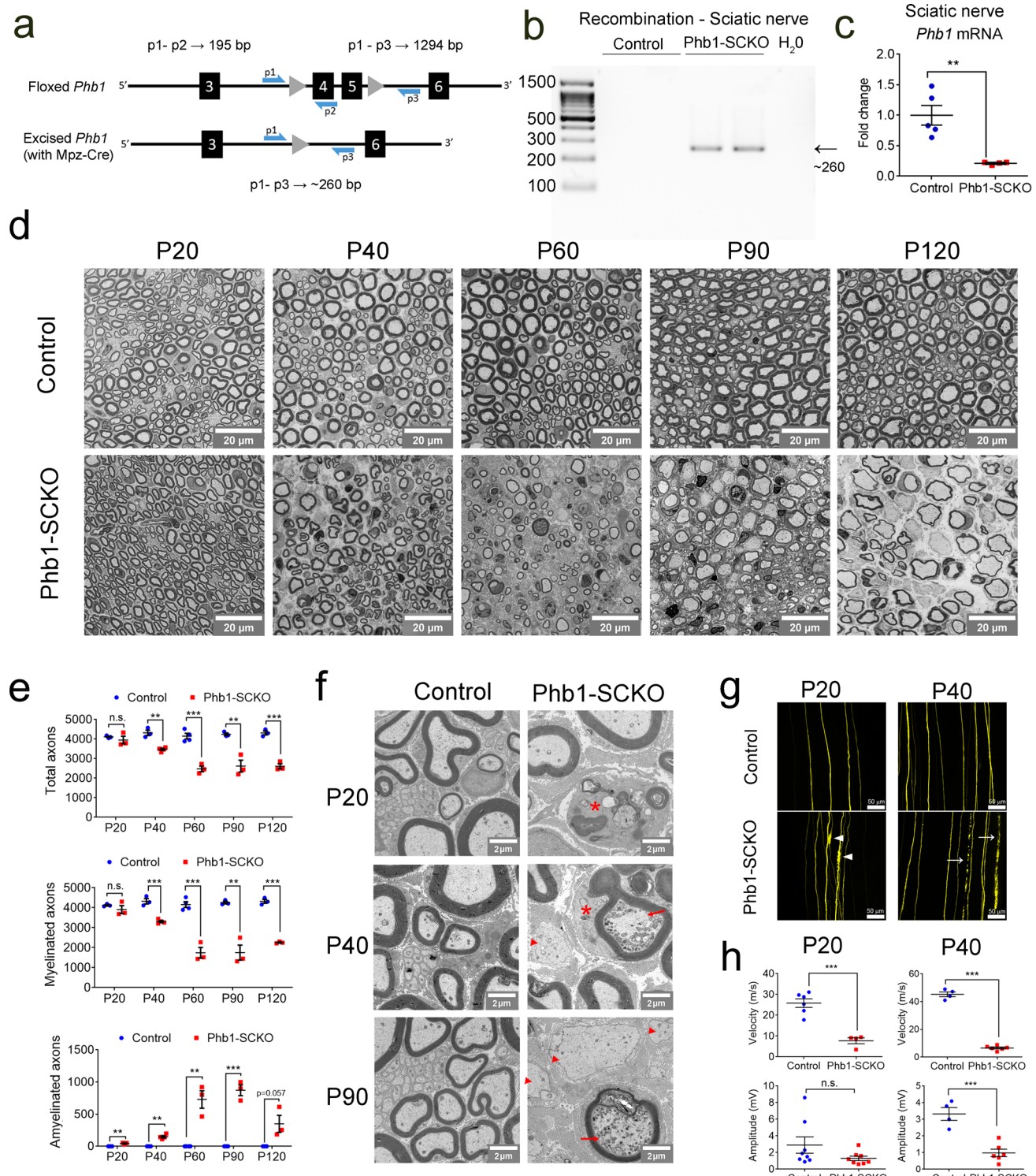

reduced (Fig. 2e, f), similarly to the reported decrease of PHB1 protein levels (Supplementary Fig. 1a and 1b). Together, these data suggest that the levels of Phb1 and Phb2 are mostly, but not completely, co-regulated, raising the possibility that these proteins may have some independent functions. These results also indicate that the effects observed in Phb1-SCKO mice could be due to the loss of both PHB1 and PHB2 proteins.

**The peripheral neuropathy caused by Phb1 deletion affects different types of nerve fibers.** Previous studies have suggested

that different aspects of SC metabolism are important to maintain homeostasis of distinct types of nerve fibers in the PNS. For example, deletion of the mitochondrial transcription factor *Tfam* in SCs results in a sensory-motor demyelinating neuropathy[6]. On the other hand, ablation of the metabolic regulator *Lkb1* in SCs causes myelination delay later followed by widespread degeneration of Remak bundles, groups of small axons ensheathed by non-myelinating SCs[15]. Therefore, we asked if specific types of fibers were more affected in Phb1-SCKO mice. The size distribution of myelinated axons in sciatic nerves, which contain both motor and sensory fibers, was similar between Phb1-SCKO

**Fig. 1 A progressive demyelination and axonal degeneration affect Phb1-SCKO animals. a** Schematic representation of the floxed *Phb1* allele. Exons 4 and 5 of the *Phb1* gene are deleted upon Cre expression. Primers p1 and p2 were used for genotyping, while primers p1 and p3 were used to evaluate recombination. Square (exon), triangle (loxP site), blue half-arrow (primer). **b** Recombination PCR on DNA isolated from sciatic nerves reveals a ~260 bp recombined band in Phb1-SCKO animals, while unrecombined DNA is too long to generate an amplicon with our PCR conditions. The experiment was repeated independently twice with identical results. **c** RT-qPCR analyses show a significant reduction in the level of *Phb1* mRNA in sciatic nerve lysates of postnatal day 20 (P20) Phb1-SCKO mice (red) compared to controls (blue). $N = 4–5$ animals per genotype. Unpaired two-tailed t-test ($t = 4.295$, df = 7, $p = 0.0036$). **d** Representative images of cross sections of sciatic nerves. $N = 3–4$ animals per genotype. **e** The number of myelinated axons per sciatic nerve is greatly reduced in Phb1-SCKO animals starting at postnatal day 40 (P40) (middle). The decline can be explained both by demyelination (bottom) and axonal degeneration, evidenced by the reduction in the total number of axons (top). $N = 3–4$ animals per genotype. Unpaired two-tailed *t*-test corrected for multiple comparisons using the Holm-Sidak method. Total axons [P20 ($t = 0.816$, df = 4, $p = 0.46$), P40 ($t = 6.508$, df = 5, $p = 0.0038$), P60 ($t = 8.08$, df = 5, $p = 0.0022$), P90 ($t = 5.257$, df = 4, $p = 0.012$), P120 ($t = 10.608$, df = 4, $p = 0.0022$)]; myelinated axons [P20 ($t = 1.053$, df = 4, $p = 0.35$), P40 ($t = 7.885$, df = 5, $p = 0.0016$), P60 ($t = 8.635$, df = 5, $p = 0.0014$), P90 ($t = 6.572$, df = 4, $p = 0.0055$), P120 ($t = 20.281$, df = 4, $p = 0.00017$)]; amyelinated axons [P20 ($t = 5.608$, df = 4, 0.0099), P40 ($t = 6.832$, df = 5, $p = 0.0041$), P60 ($t = 6.371$, df = 5, $p = 0.0042$), P90 ($t = 10.346$, df = 4, $p = 0.0024$), P120 ($t = 2.643$, df = 4, $p = 0.057$)]. **f** Representative electron micrographs demonstrating the presence of degenerating axons (arrows), amyelinated/demyelinated axons (arrowheads) and SCs degrading their own myelin (star). $N = 3$ animals per genotype. **g** Sparse labeling of axons in the tibial nerve using the Thy1-YFP reporter mouse indicates the presence of axonal swelling at P20 (arrowheads) and axon fragmentation at P40 (arrows). $N = 3–4$ animals per genotype. **h** A functional decline is detected in Phb1-SCKO animals by electrophysiological measurements, with a reduction in nerve conduction velocity as early as P20 and decreased CMAP amplitude at P40. $N = 4–8$ animals per genotype. Unpaired two-tailed *t*-test. Velocity [P20 ($t = 6.387$, df = 8, $p = 0.0002$), P40 ($t = 23.76$, df = 8, $p < 0.0001$)]; Amplitude [P20 ($t = 1.574$, df = 14, $p = 0.14$), P40 ($t = 5.635$, df = 8, $p = 0.0005$)]. Data are presented as mean ± SEM. *$p < 0.05$, **$p < 0.01$, ***$p < 0.001$. n.s. = non-significant.

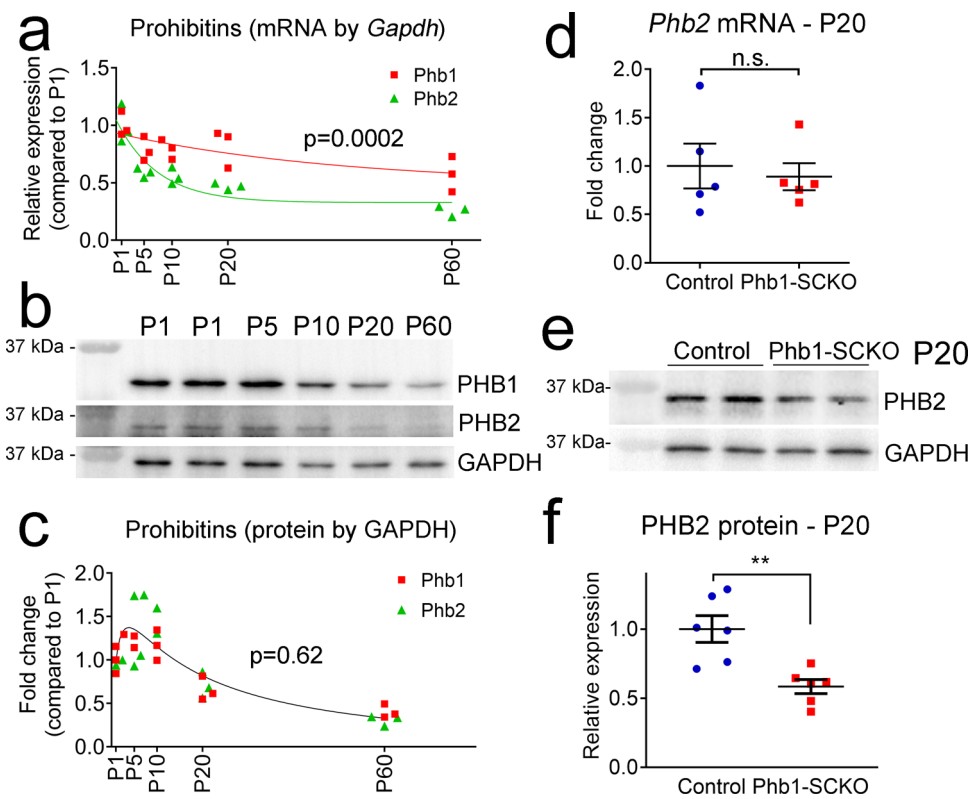

**Fig. 2 Levels of Phb1 and Phb2 are mostly but not completely interdependent. a** Developmental trajectory of *Phb1* (red) and *Phb2* (green) mRNA in sciatic nerves. $N = 3$ animals per time point. Non-linear regression modeled from a one-phase decay function followed by extra sum-of-squares F test; $F_{(3, 24)} = 10.24$. **b** Western blots of sciatic nerve lysates at the indicated ages. **c** Quantification of the changes in expression of PHB1 and PHB2 in sciatic nerves over time. $N = 3$ samples per time point. Samples at postnatal day 1 (P1) and P5 were pooled from 4 and 2 animals, respectively. Non-linear regression modeled from a lognormal function followed by extra sum-of-squares F test; $F_{(3, 24)} = 0.6188$. **d** RT-qPCR from P20 sciatic nerves show that *Phb2* mRNA levels are not altered by deletion of *Phb1*. $N = 5$ animals per genotype. Unpaired two-tailed *t*-test; $t = 0.4075$, df = 8. **e** Western blot from sciatic nerve lysates of P20 Phb1-SCKO mice illustrating the reduction of PHB2 levels. **f** Quantification of (**e**). $N = 6$ animals per genotype. Unpaired two-tailed *t*-test; $t = 3.825$, df = 10, $p = 0.0033$. Data are presented as mean ± SEM. **$p < 0.01$. n.s. non-significant.

and controls (Supplementary Fig. 2a and b), suggesting that demyelination equally affects fibers of all calibers. Demyelinated axons showed reduced axon caliber compared to myelinated axons in Phb1-SCKO animals (Supplementary Fig. 2c and d), in line with previous findings that demyelination results in

reduction of axon caliber due to decreased neurofilament phosphorylation[16], which, in turn, may be a consequence of loss of myelin and trophic support from SCs.

We also directly investigated if peripheral neuropathy was prominent in both the motor and sensory compartments by

comparing the primarily sensory saphenous nerve and the motor branch of the femoral nerve (Supplementary Fig. 3a). Phb1-SCKO animals show a reduction of myelinated fibers and the presence of demyelinated fibers in both the femoral motor and the saphenous nerve (Supplementary Fig. 3b and c, respectively). Therefore, Phb1 deletion from SCs causes sensory-motor peripheral neuropathy.

Last, we examined the effect of Phb1 ablation on Remak SCs, which surround non-myelinated small-caliber axons. At P20, Phb1-SCKO mice had a typical density and morphology of Remak bundles, which displayed normal organization with well-ensheathed axons of the correct number and size (Supplementary Fig. 4). However, at P40, several Remak SCs of Phb1-SCKO mice seemed to have retracted their processes, resulting in many axons directly abutting each other (Supplementary Fig. 4). This phenomenon was somewhat lessened at P90, however, at this age, Remak bundles of Phb1-SCKO mice were fragmented, which resulted in a reduced number of axons per Remak bundle and an increased density of Remak bundles per nerve (Supplementary Fig. 4b). In addition, at P90, Remak bundles of Phb1-SCKO mice tended to contain abnormally large axons. In combination, these results may reflect an attempt of the Remak SCs to respond to the ongoing demyelination by: (1) transdifferentiating to a SC phenotype that promotes nerve repair and; (2) ensheathing some small, demyelinated axons. Alternatively, Remak SCs of Phb1-SCKO mice may simply be dysfunctional, causing fragmentation of the Remak bundle and axonal swelling.

**Macrophage infiltration, ERK activation, and SC death are unlikely to be causes of the neuropathy in Phb1-SCKO mice.** For all the subsequent analyses, we focused on three time points: P20, P40, and P90, which represent beginning, middle and late stages of peripheral neuropathy, respectively. It is a reasonable assumption that alterations that are already present at P20 are more likely to be causal for the phenotype, while alterations that appear at the P40 or P90 time points are most likely secondary phenomena that originate as a consequence of the pathology.

Given that prohibitins are required for activation of the Raf-MEK-ERK pathway by Ras[17], and that this is an important pathway for myelination[18] and demyelination[19], we first asked whether Phb1 deletion in SCs affected ERK1/2 expression or phosphorylation. Surprisingly, we only found minor changes in p-ERK1/2 and total ERK1/2 at P40 (Supplementary Fig. 5a and b).

A common feature of peripheral neuropathies is the presence of macrophages, which infiltrate the nerves to phagocytose myelin and cellular debris. Occasionally, macrophages can be involved with the initiation of peripheral neuropathies and cause demyelination[20]. Thus, we asked if macrophages were present in the nerves of Phb1-SCKO mice before demyelination. An elevated number of macrophages is detectable in P40 and P90 Phb1-SCKO animals in comparison to controls (Supplementary Fig. 5b), but there are no differences at P20 (Supplementary Fig. 5c and d). The timing of macrophage infiltration (P40) coincides with the elevation of ERK1/2 phosphorylation (Supplementary Fig. 5b) and of Mcp1 upregulation (Supplementary Fig. 5e), a chemokine previously shown to be secreted by SCs and fibroblasts downstream of ERK signaling to promote macrophage recruitment[21]. Therefore, it is possible that all these events are linked. Nonetheless, they are most likely a consequence, rather than a cause, of demyelination.

Prohibitins are often necessary to maintain cell survival and support cell proliferation[22]. In fact, knockdown of either Phb1 or Phb2 in isolated primary rat SCs leads to cell death[8]. Thus, we investigated whether SC death or proliferation were altered in vivo in nerves of Phb1-SCKO mice. At P20, there were no

differences between groups in TUNEL assay and staining for the mitotic marker phosphorylated Histone 3 (p-H3) (Supplementary Fig. 6a and b). However, nerves of Phb1-SCKO animals showed a slight increase in both TUNEL+ and p-H3+ cells at P40 and P90 (Supplementary Fig. 6a and b). We further confirmed that TUNEL+ and proliferating cells were SCs by co-staining with SOX10 (Supplementary Fig. 6c and d). Moreover, using the broader proliferation marker, Ki-67, we found increased proliferation in SCs of Phb1-SCKO mice at P20 and P40 (Supplementary Fig. 6e). Nonetheless, most of the changes in cell survival and proliferation were tardy and modest and thus unlikely to be sufficient and timely to cause the phenotype. In addition, the balance between cell death and cell division seems to be maintained, since nerves of Phb1-SCKO contain normal quantities of SCs at P20 and P40 (Supplementary Fig. 6f).

**Ablation of prohibitin 1 causes accumulation of damage in mitochondria.** Prohibitins are mainly localized in the mitochondria, where they are essential for their structure and function. Prohibitins are important for many mitochondrial functions, including fusion, cristae morphogenesis[23], mitochondrial DNA (mtDNA) maintenance[24], stabilization of respiratory complexes[25,26], and prevention of reactive oxygen species (ROS) production[27]. Thus, we expected that mitochondria would be primarily and severely affected in Phb1-SCKO animals. Analyses by electron microscopy revealed aberrant mitochondrial morphology (Fig. 3a), with SCs of Phb1-SCKO mice showing an increase in the average mitochondrial perimeter at all evaluated time points (Fig. 3b–d). This result suggests the presence of mitochondrial swelling, a finding typical of dysfunctional mitochondria associated with several pathological conditions[28].

To investigate SC mitochondria in more detail, we crossed Phb1-SCKO animals to PhAM mice[29], which express a flox-STOP mitochondrial fluorescent reporter. The Mpz-Cre mediated recombination of PhAM exclusively in SCs allows for a selective evaluation of fluorescent SC mitochondria without confounding results from a large number of mitochondria present in axons. We analyzed confocal images of teased sciatic nerve fibers using an automated routine for ImageJ, which allowed quantification of mitochondrial volume in different compartments of the SC. In myelinating SCs, mitochondria are enriched around the endoplasmic reticulum (ER) near the SC cell body, adjacent to nodes of Ranvier in regions called the paranode and juxtaparanode, and in longitudinal cytoplasmic channels between the external surface of myelin and the SC plasma membrane (Cajal bands) (Fig. 3e). At P20, larger-sized mitochondria are overrepresented around the ER and in Cajal bands of Phb1-SCKO animals compared to controls, while mitochondria in juxtaparanodes show a shift toward smaller sizes (Fig. 3f, quantified in Supplementary Fig. 7). We also identified a trend toward increased mitochondrial numbers in Cajal bands of Phb1-SCKO mice (Supplementary Fig. 7b). At P40, aberrant SC mitochondrial patterning in Phb1-SCKO mice persists, with altered size distribution of mitochondria in the vicinity of Cajal bands, the nodes/paranodes, and in Remak SCs (Supplementary Fig. 8). Strikingly, at P40, many myelinating SCs of Phb1-SCKO mice displayed an almost complete absence of PhAM signal in portions of the SC away from the cell body (Fig. 3g, arrows). This phenomenon affected about 20% of all myelinating SCs of P40 Phb1-SCKO animals (Fig. 3h). In addition, other mitochondrial markers, such as HSPD1 and TOM20 were also severely reduced in SCs in which PhAM was undetectable (Supplementary Fig. 9). This likely reflects the amplification of the early mitochondrial dysfunction seen at P20 and consequent fragmentation and elimination of damaged mitochondria. In line with this hypothesis, there is

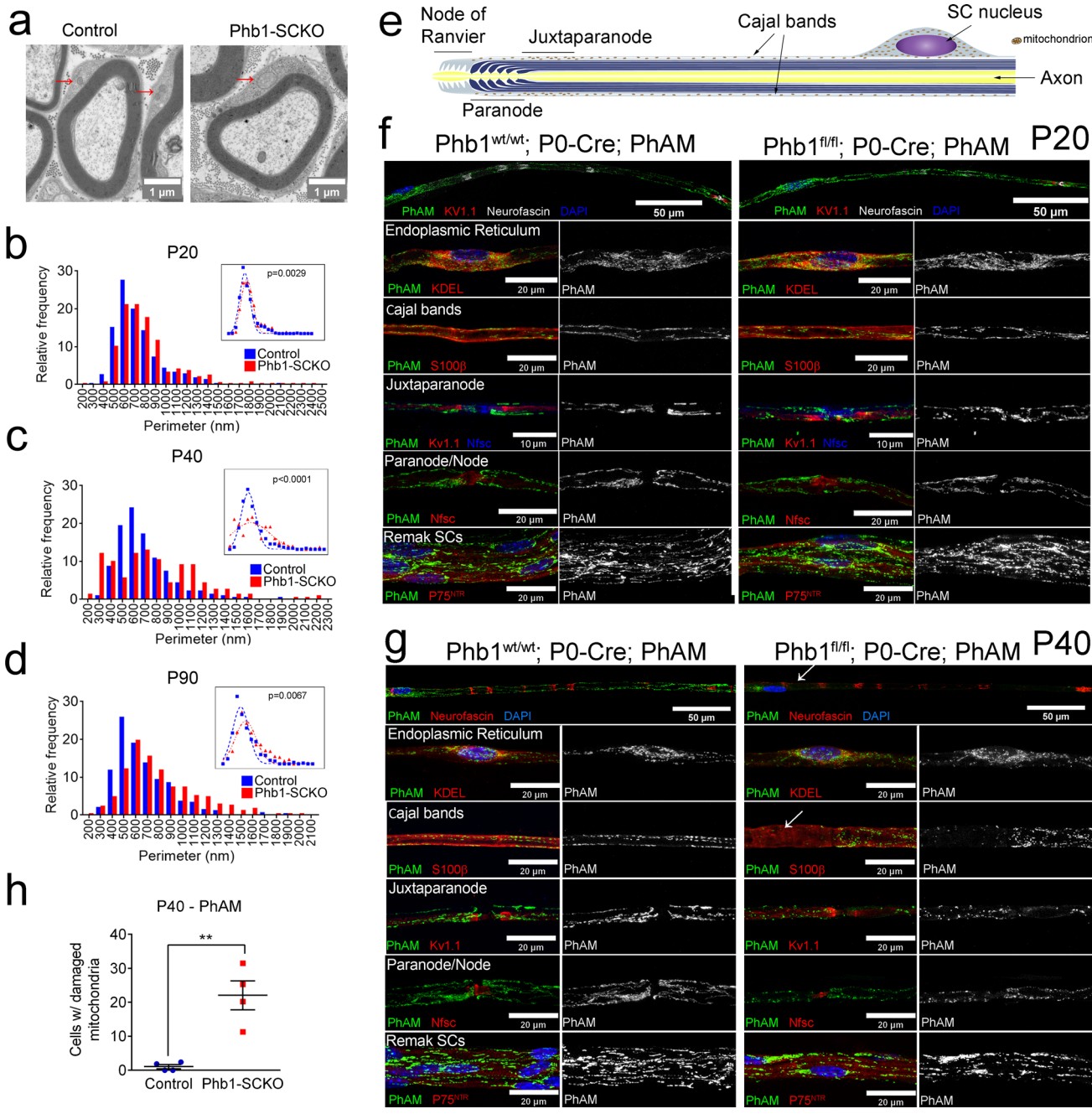

**Fig. 3 Ablation of Prohibitin 1 in Schwann cells results in altered mitochondrial morphology. a** Representative electron micrographs highlighting the enlargement of mitochondria in sciatic nerves of Phb1-SCKO animals at postnatal day 20 (P20) (arrows). $N = 3$ animals per genotype. (**b-d**) Mitochondria in SCs of PHB1-SCKO mice (red) have a larger perimeter compared to mitochondria of control animals (blue) at P20 (**b**), P40 (**c**), and P90 (**d**). At P40, there is also a population of mitochondria that has a reduced perimeter, suggesting mitochondrial fragmentation. This population is lost at P90, suggesting that the fragmented mitochondria disappear. $N = 3$ animals per genotype; at least 100 mitochondria from each animal were evaluated. Insets: non-linear regression using a Gaussian curve followed by extra sum-of-squares $F$ test [$F(3,42)$ P20 = 5.482, $F(3,38)$ P40 = 19.48, $F(3,34)$ P90 = 4.813). **e** Schematic representation of the distribution of mitochondria in a myelinating SC. **f–g** Confocal z-projections of teased fibers of sciatic nerves of Phb1-SCKO mice and controls illustrating the morphology of Schwann cell mitochondria as labeled by the PhAM reporter (green) near different cellular structures (red). DAPI is indicated in blue. **f** At P20, there are changes in mitochondrial size. $N = 4$ animals per genotype. **g** At P40, some cells lack PhAM expression away from the cell body. $N = 3$ animals per genotype. **h** PhAM is not detectable in about 20% of the myelin internodes of Phb1-SCKO animals at P40. $N = 4$ animals per genotype. Unpaired two-tailed $t$-test ($t = 4.866$, df = 6, $p = 0.0028$). Data are presented as mean ± SEM. **$p < 0.01$.

progressive depletion of mtDNA from sciatic nerves of Phb1-SCKO animals (Fig. 4a) and consequent reduction in levels of transcripts encoded from the mtDNA at P40 in the same tissue (Supplementary Fig. 10). In addition, sciatic nerves of P40 Phb1-SCKO mice showed a reduction in mRNA levels of *Tfam* and a trend toward reduction of *PGC1α*, two genomically encoded

transcriptional regulators essential for mitochondrial physiology (Supplementary Fig. 10).

Mitochondria are dynamic organelles, constantly undergoing division (fission) and fusion and being transported to specific cellular locations[30]. Defects in mitochondrial dynamics could contribute to the mosaic pattern of mitochondrial loss in SCs of

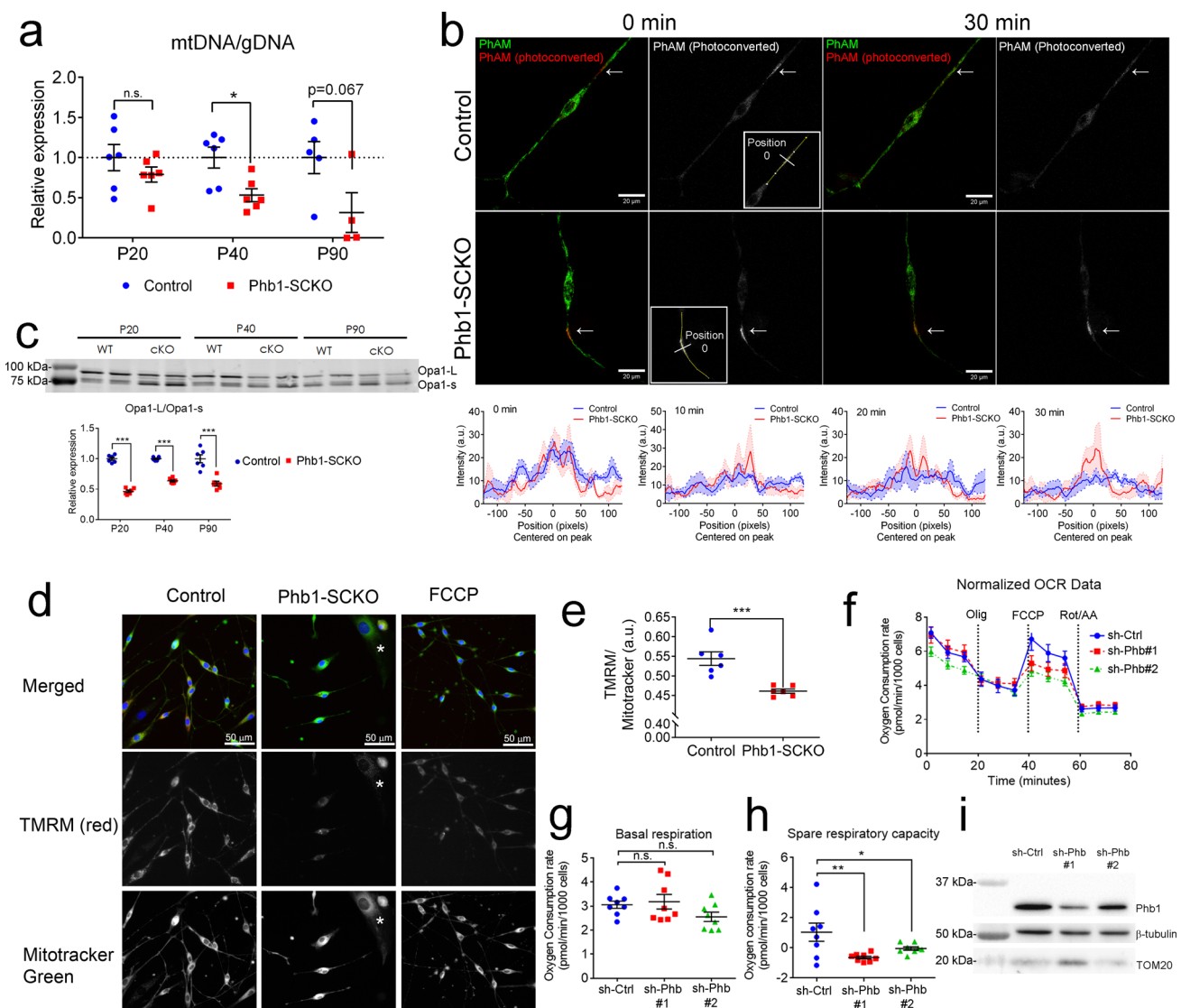

**Fig. 4 Mitochondria of Phb1-SCKO are dysfunctional. a** The mitochondrial DNA (mtDNA) content is decreased in Phb1-SCKO mice (red) compared to controls (blue) starting at postnatal day 40 (P40). $N = 4$–6 animals per genotype. Unpaired two-tailed $t$-test corrected for multiple comparisons using the Holm-Sidak method [P20 ($t = 1.114$, df $= 10$, $p = 0.29$), P40 ($t = 3.068$, df $= 10$, $p = 0.012$), P90 ($t = 2.174$, df $= 7$, $p = 0.066$)]. **b** Mitochondrial dynamics is affected by deletion of *Phb1* in primary mouse SCs. Top: PhAM fluorescence was photoconverted using a focal laser stimulation and dynamics of the photoconverted mitochondria (arrows) were observed for 30 min. While the PhAM (photoconverted) signal quickly dissipated in the control SCs, it remained stagnant in Phb1-SCKO SCs. Inset: position 0 represents the center of the stimulated area. Bottom: quantification of PhAM (photoconverted) signal around the stimulated area at different time points. Mean (line) and SEM (shaded area) of the signal is reported. A.u. = arbitrary units. $N = 6$–7 cells per genotype. **c** Processing of the mitochondrial fusion protein Opa1 is increased in Phb1-SCKO mice. $N = 6$ animals per genotype. Unpaired two-tailed $t$-test [P20 ($t = 16.664$, df $= 10$, $p < 0.000001$), P40 ($t = 17.015$, df $= 10$, $p < 0.000001$), P90 ($t = 5.335$, df $= 10$, $p = 0.00033$)]. **d** Ablation of Phb1 leads to reduced mitochondrial membrane potential in primary SCs of P40 PHB1-SCKO mice as compared to controls. **e** Quantification of (**d**). $N = 6$ wells per genotype. Unpaired two-tailed $t$-test ($t = 4.583$, df $= 10$, $p = 0.001$). Asterisk: fibroblast. **f**–**h** In primary rat SCs, Seahorse analyses indicate that *Phb1* knockdown with either sh-Phb #1 (red) or sh-Phb #2 (green) does not change basal mitochondrial respiration (One-way ANOVA. F (2, 21) $= 2.195$, $p = 0.1363$) (**g**), but impairs the spare respiratory capacity (**h**). Timing of injection of oligomycin (Olig), FCCP, and rotenone (Rot) + antimycin A (AA) are indicated in (**f**). **i** Western blot illustrating the reduction in levels of PHB1 upon treatment with shRNAs (63% and 21% reduction in PHB1/β-tubulin ratio for sh-Phb #1 and sh-Phb #2, respectively). $N = 8$ wells per condition. One-way ANOVA. F (2,21) $= 5.594$, $p = 0.0113$; $p_{sh-Phb1\#1} = 0.0068$; $p_{sh-Phb1\#2} = 0.0456$. Data are presented as mean ± SEM. *$p < 0.05$, **$p < 0.01$, ***$p < 0.001$. n.s. non-significant.

Phb1-SCKO mice by leading to the progressive accumulation and amplification of mitochondrial derangements in specific SCs. To investigate if mitochondrial dynamics were altered by deletion of *Phb1* in SCs, we performed live imaging of SCs isolated from Phb1$^{wt/wt}$; P0-Cre; PhAM (Control) and Phb1$^{fl/fl}$; P0-Cre; PhAM (Phb1-SCKO) animals. We took advantage of the photoconvertible nature of the Dendra2 fluorophore in the PhAM mice. When stimulated by visible blue or UV-violet light, Dendra2

converts from a green to a red fluorescent state. We exposed a focal region of the SCs to the 405 nm confocal laser to promote photoconversion of PhAM and then monitored cells for 30 min, tracking the photoconverted mitochondria. In control SCs, photoconverted (red) mitochondria dispersed quickly and no peak in the red signal is evident at the end of the experiment (Fig. 4b). On the other hand, red SC mitochondria are still present as a group after 30 min in Phb1-SCKO mice (Fig. 4b and

Supplementary Videos 2 and 3). This indicates that deletion of *Phb1* in SCs leads to impaired mitochondrial dynamics. Defective mitochondrial dynamics could be a consequence of dysfunctional Opa1, a dynamin-related GTPase involved in the fusion of the inner mitochondrial membrane. Deletion of prohibitins has been previously reported to trigger proteolytic cleavage of Opa1, impairing its function[23]. In agreement with this hypothesis, western blots from sciatic nerve lysates show increased proteolytic processing of Opa1 in Phb1-SCKO mice (Fig. 4c).

Next, we asked whether these mitochondrial changes affected mitochondrial physiology and function. We evaluated mitochondrial membrane potential using TMRM in SCs isolated from P40 control and Phb1-SCKO animals. Ablation of Phb1 results in a significant reduction in TMRM fluorescence, suggesting that mitochondria of Phb1-SCKO mice are depolarized (Figs. 4d and e). Since the mitochondrial membrane potential relates to the cell's capacity to make ATP through oxidative phosphorylation, we also evaluated mitochondrial respiration using the Seahorse Extracellular Flux Analyzer. As SCs of Phb1-SCKO mice have impaired survival in vitro and a relatively large number of cells were required for this analysis, we instead acutely knocked down *Phb1* in primary rat SCs using shRNA. Our analyses revealed normal basal respiration in *Phb1*-knockdown SCs, but a significant decrease in the spare respiratory capacity compared to cells treated with sh-Control (Fig. 4f–i). These results indicate that SCs lacking PHB1 may be unable to appropriately respond to changes in metabolic demand.

A common consequence of inefficient respiration in a situation of stress is the production of ROS by mitochondria. However, we did not detect any difference in lipoperoxidation or protein oxidation between nerves of Phb1-SCKO mice and controls (Supplementary Fig. 11), suggesting that, at least at the time points examined, oxidative stress may not be present.

Phb2 has recently been described as a mitophagy receptor at the inner mitochondrial membrane[31]. Although it is not clear if Phb1 also participates in mitophagy, we postulated that PHB1-ablated SCs may accumulate damaged mitochondria because they may be unable to perform mitophagy. We thus analyzed the capacity of PHB1-ablated SCs to carry out mitophagy. For these analyses, we made use of a retrovirus system to deliver mt-mKeima, a mitochondrially targeted pH-sensitive fluorescent protein[32]. When in the mitochondria (which has a pH of about 7.8), the mt-mKeima excitation peak is at 440 nm. However, when mitochondria are targeted to degradation in the lysosomes (therefore reducing the pH), the excitation peak of mt-mKeima shifts to 586 nm. Thus, a ratio of the mt-mKeima fluorescence at ~586 nm over the fluorescence at ~440 nm can be used as a proxy for the degree of mitochondrial degradation in the lysosomes. Using this method, we found that silencing of Phb1 by shRNA does not change the ability of SCs to perform mitophagy (Supplementary Fig. 12).

We next tested if there is an association between mitochondrial damage and demyelination in Phb1-SCKO mice. To this end, we analyzed teased fibers from P40 Phb1[wt/wt]; P0-Cre; PhAM (Control) and Phb1[fl/fl]; P0-Cre; PhAM (Phb1-SCKO) animals. In Phb1-SCKO animals, fibers with undetectable PhAM were overrepresented among fibers containing cytoplasmic myelin inclusions (myelin ovoids), suggesting an association between mitochondrial damage and demyelination (Supplementary Fig. 13).

Taken together, our results indicate that mitochondria in SCs of Phb1-SCKO mice are severely impaired starting at P20. In addition, we show evidence that mitochondrial dysfunction and demyelination are linked and that the mitochondrial damage is accumulating, possibly because Phb1-SCKO mice have abnormal mitochondrial dynamics.

**Deletion of *Phb1* activates a mitochondrial stress response.** Although mitochondria have their own DNA, they still depend on genomic DNA to synthetize most of their proteins (only 13 out of more than 1200 mitochondrial proteins are encoded by mtDNA)[33]. Thus, in order to maintain homeostasis, there is a need for bidirectional mitonuclear communication[34]. In a similar fashion, when mitochondria are under stress, cells respond with a coordinated attempt to mitigate potential damage. In mammals, this normally involves activation of the ISR, a general stress pathway working to reduce overall protein synthesis and favor the expression of stress-response genes[35]. A previous report by Viader et al.[7] showed that deletion of the mitochondrial transcription factor *Tfam* in SCs results in activation of the ISR, which they postulated to be a maladaptive mechanism, although this was not directly tested. Thus, we asked whether *Phb1* deletion in SCs triggers the ISR by assessing the levels of phosphorylated eIF2α, the hallmark of the ISR. Indeed, we found that nerves of Phb1-SCKO mice showed early (P20) and continually elevated levels of phosphorylated eIF2α in comparison to littermate controls (Fig. 5a).

We then sought to investigate the downstream events in this mitochondrial stress response. Classically, mitochondrial dysfunction induces the mitochondrial unfolded protein response (UPR[mt]), which results in upregulation of a set of genes including mitochondrial chaperones (such as *Hspd1* and *Hspe1*, also known as *Hsp60* and *Hsp10*, respectively) and proteases (such as *Clpp*). Thus, we probed Phb1-SCKO animals for the activation of UPR[mt]. Surprisingly, we did not detect elevation of any UPR[mt] effector at the protein or RNA level (Fig. 5b–d). A recent publication implicated ATF4 in the mammalian mitochondrial stress response and characterized the molecular signature of this pathway[36]. Therefore, we assessed this pathway in nerves of Phb1-SCKO mice. Even though *Atf4* expression itself is not altered by deletion of *Phb1* in SCs, four out of the five analyzed transcripts regulated by ATF4 are upregulated in nerves of Phb1-SCKO mice (Fig. 5e): *Asns* (asparagine synthetase); *Chac1* (cation transport regulator-like protein 1); *Pck2* (phosphoenolpyruvate carboxykinase 2) and *Dddit3* (DNA damage-inducible transcript 3; also known as *Chop*). Importantly, phosphorylation of eIF2α in this context does not seem to be trigged by activation of PERK kinase in the ER, since its phosphorylation is not increased (Supplementary Fig. 14a and b). However, Phb1-SCKO mice do show elevated levels of the HSPA chaperone (also known as Bip) (Supplementary Fig. 14c) and upregulated alternative splicing of *Xbp1* (Supplementary Fig. 14d), two markers commonly associated with the unfolded protein response in the ER (UPR[ER]). This suggests that mitochondrial dysfunction caused by Phb1-SCKO indirectly leads to ER stress.

The ER is an organelle involved in lipid synthesis, while mitochondria take part in beta-oxidation, the process of breakdown of fatty acids. Therefore, it is possible that the balance of lipid metabolism is altered in Phb1-SCKO mice. If lipid oxidation occurs disproportionally to lipid synthesis, myelin maintenance could be affected due to the depletion of important myelin lipids. This mechanism has been previously proposed to underlie the peripheral neuropathy caused by deletion of *Tfam* in SCs[7]. Therefore, we evaluated the expression and phosphorylation of Acetyl-CoA carboxylase (ACC). The ACC enzyme is responsible for the production of malonyl-CoA, the substrate for biosynthesis of fatty acids and an inhibitor of beta-oxidation. At P20, nerves of Phb1-SCKO mice showed increased inhibitory phosphorylation of ACC (Fig. 6a–d). Moreover, the expression of many genes involved with lipid synthesis is severely reduced (Fig. 6e, f) at both P20 and P40, suggesting that reduction of lipid biosynthesis might

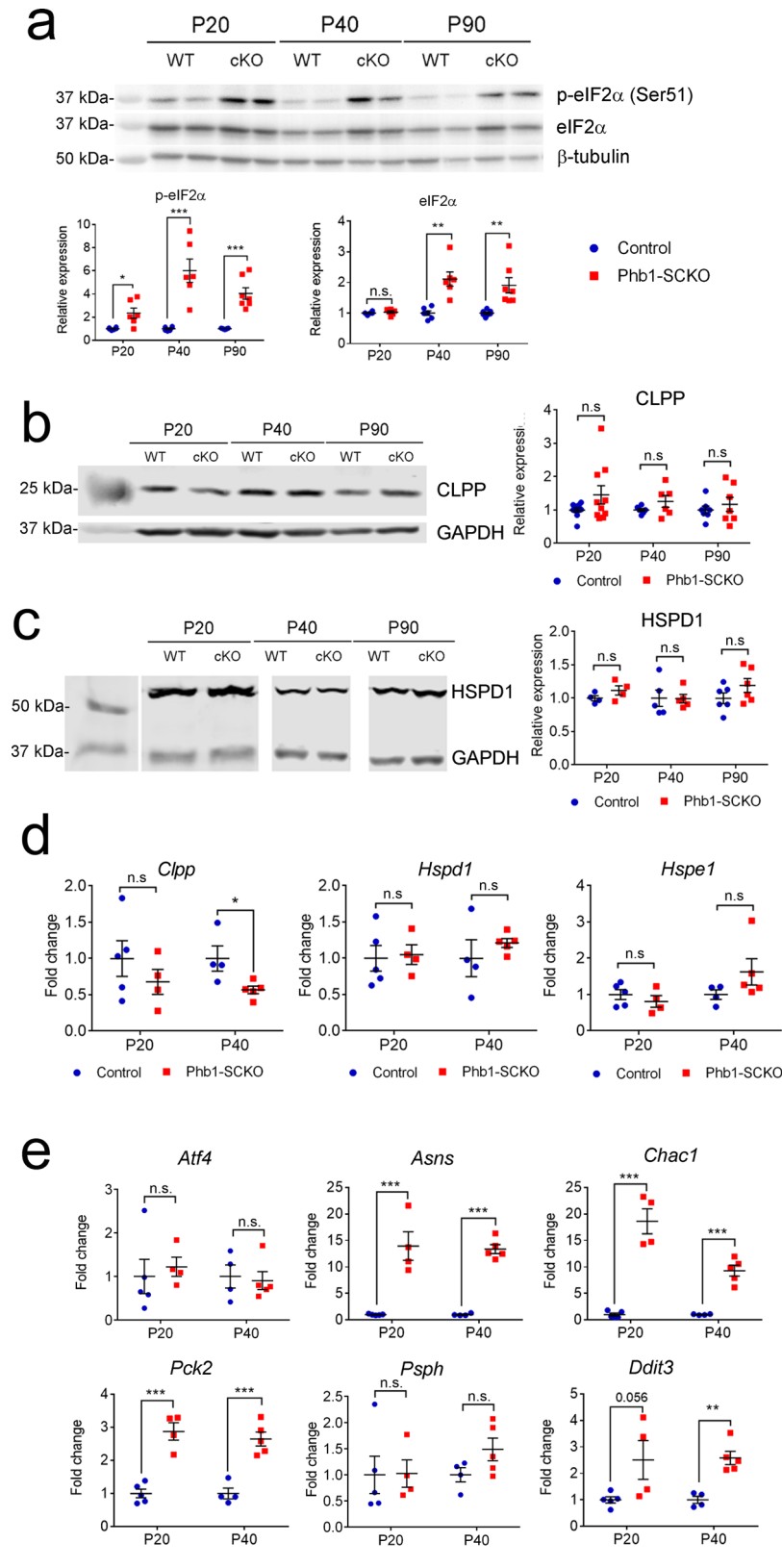

be a common finding in neuropathies in which SC mitochondria are damaged.

In summary, ablation of *Phb1* in SCs leads not only to a mitochondrial stress response involving the ISR, but also to a broader cellular response, affecting the ER and causing reduced expression of enzymes involved in lipid biosynthesis.

**The mitochondrial stress response is beneficial to Phb1-SCKO mice**. Given the dramatic implications of the ISR for cells, such as a widespread inhibition of translation, we aimed to test if continuous activation of the ISR was maladaptive in the context of mitochondrial dysfunction in SCs, as suggested by Viader et al.[7]. To evaluate the role of ISR in Phb1-SCKO mice, we treated

**Fig. 5 Deletion of *Phb1* in SCs triggers a mitochondrial stress response. a** p-eIF2α is upregulated in sciatic nerve lysates of PHB1-SCKO mice (red) compared to controls (blue), indicating activation of the ISR. $N = 6$–7 animals per genotype. Unpaired two-tailed $t$-test. p-eIF2α [P20 ($t = 3.068$, df $= 10$, $p = 0.013$), P40 ($t = 4.948$, df $= 10$, $p = 0.0019$), P90 ($t = 6.088$, df $= 12$, $p = 0.0029$)]; eIF2α [P20 ($t = 0.759$, df $= 10$, $p = 0.091$), P40 ($t = 4.532$, df $= 10$, $p = 0.023$), P90 ($t = 3.582$, df $= 12$, $p = 0.018$)]. **b** Representative western blot for CLPP, a protease involved in the UPR$^{mt}$ response (left) and quantification of relative expression levels at different time points (right). $N = 6$-10 animals per genotype. Unpaired two-tailed $t$-test [P20 ($t = 1.616$, df $= 18$, $p = 0.12$), P40 ($t = 1.416$, df $= 10$, $p = 0.19$), P90 ($t = 0.696$, df $= 12$, $p = 0.5$)]. **c** Representative immunoblot for HSPD1, a chaperone participating in the UPR$^{mt}$ cascade (left) and quantification of relative expression levels at different time points (right). $N = 4$-5 animals per genotype. Unpaired two-tailed $t$-test [P20 ($t = 1.46$, df $= 6$, $p = 0.19$), P40 ($t = 0.057$, df $= 8$, $p = 0.96$), P90 ($t = 1.46$, df $= 10$, $p = 0.17$)]. **d** RT-qPCR analysis of gene expression of *Clpp*, *Hspd1* and *Hspe1* (*Hsp10*). $N = 4$-5 animals per genotype. Unpaired two-tailed $t$-test *Clpp* [P20 ($t = 1.013$, df $= 7$, $p = 0.34$), P40 ($t = 2.642$, df $= 7$, $p = 0.033$)]; *Hspd1* [P20 ($t = 0.217$, df $= 7$, $p = 0.83$), P40 ($t = 0.9$, df $= 7$, $p = 0.4$)]; *Hspe1* [P20 ($t = 0.899$, df $= 7$, $p = 0.4$), P40 ($t = 1.472$, df $= 7$, $p = 0.18$)]. **e** The mitochondrial stress response involves ATF4, as suggested by the upregulation of its targets: *Asgn* (aspargine synthetase), *Chac1* (cation transport regulator-like protein 1), *Pck2* (phosphoenolpyruvate carboxykinase 2), *Dddit3* (DNA damage-inducible transcript 3), *Psph* (phosphoserine phosphatase). $N = 4$-5 animals per genotype. Unpaired two-tailed $t$-test. *Atf4* [P20 ($t = 0.456$, df $= 7$, $p = 0.66$), P40 ($t = 0.288$, df $= 7$, $p = 0.78$)]; *Asns* [P20 ($t = 12.772$, df $= 7$, $p = 0.00095$), P40 ($t = 5.459$, df $= 7$, $0.000004$)]; *Chac1* [P20 ($t = 8.361$, df $= 7$, $0.000069$), P40 ($t = 7.134$, df $= 7$, $0.00019$)]; *Pck2* [P20 ($t = 6.84$, df $= 7$, $p = 0.00024$), P40 ($t = 5.897$, df $= 7$, $p = 0.0006$)]; *Psph* [P20 ($t = 0.051$, df $= 7$, $p = 0.96$), P40 ($t = 1.79$, df $= 7$, $p = 0.12$)]; *Ddit3* [P20 ($t = 2.292$, df $= 7$, $p = 0.056$), P40 ($t = 5.113$, df $= 7$, $p = 0.0014$)]. Data are presented as mean ± SEM. *$p < 0.05$, **$p < 0.01$, ***$p < 0.001$. n.s. non-significant.

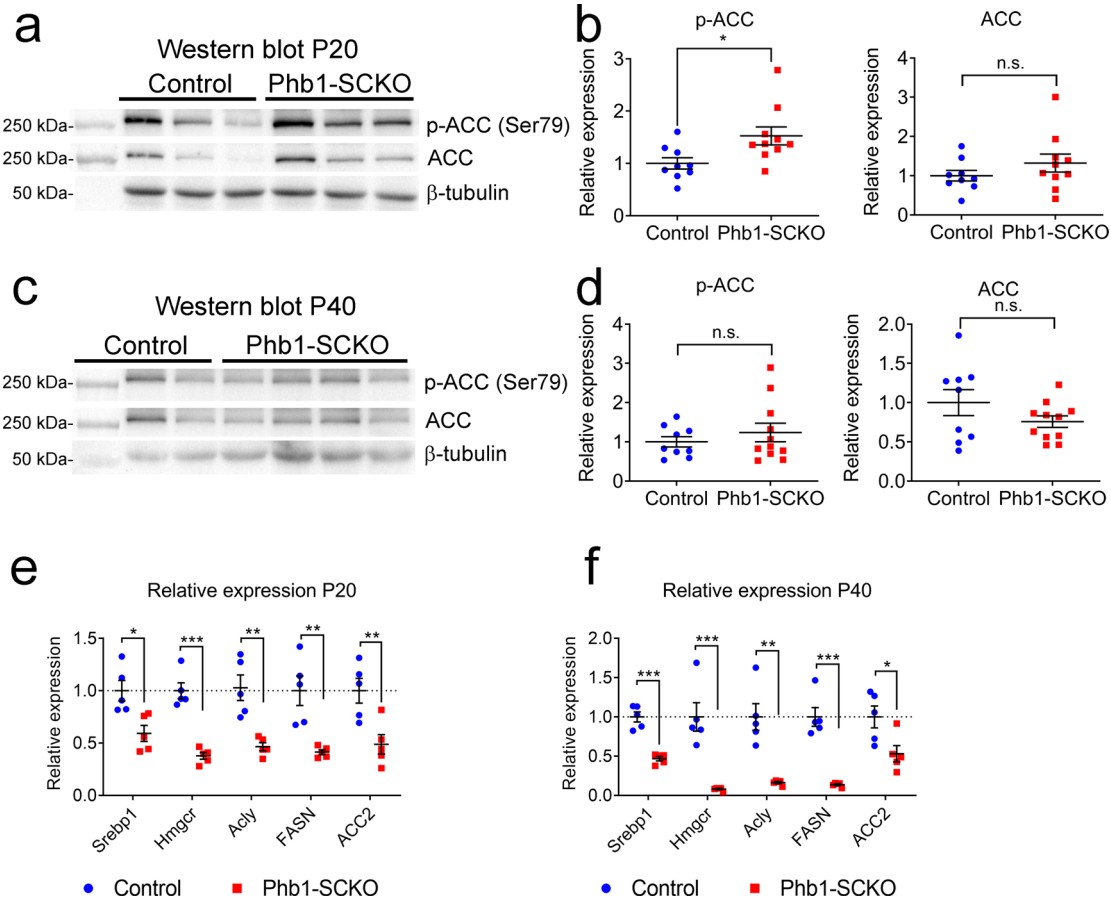

**Fig. 6 Deletion of Phb1 affects lipid metabolism.** Western blot (**a**) and quantification (**b**) of Acetyl-CoA carboxylase (ACC) and phosphorylated ACC (p-ACC) expression at P20. $N = 6$-7 animals per genotype. Unpaired two-tailed $t$-test [p-ACC ($t = 0.4627$, df $= 11$, $p = 0.021$), ACC ($t = 1.355$, df $= 11$, $p = 0.26$)]. Western blot (**c**) and quantification (**d**) of ACC and p-ACC expression at P40. $N = 6$-8 animals per genotype [p-ACC ($t = 0.5447$, df $= 12$, $p = 0.42$), ACC ($t = 1.153$, df $= 12$, $p = 0.17$)]. Unpaired two-tailed $t$-test. By RT-qPCR, we identified a significant downregulation of many enzymes involved with lipid biosynthesis at both P20 (**e**) and P40 (**f**): sterol regulatory element-binding protein 1 (*Srebp1*), 3-hydroxy-3-methylglutaryl-CoA reductase (*Hmgcr*), ATP citrate lyase (*Acly*), fatty acid synthase (*FASN*), acetyl-CoA carboxylase 2 (*ACC2*), $N = 5$ animals per genotype. Unpaired two-tailed $t$-test P20 [*Srebp1* ($t = 3.26$, df $= 8$, $p = 0.012$), *Hmgcr* ($t = 7.63$, df $= 8$, $p = 0.000061$), *Acly* ($t = 4.418$, df $= 8$, $p = 0.0022$), *FASN* ($t = 4.109$, df $= 8$, $p = 0.0034$), *ACC2* ($t = 3.408$, df $= 8$, $p = 0.0092$)]; P40 [*Srebp1* ($t = 7.551$, df $= 8$, $p = 0.000066$), *Hmgcr* ($t = 5.091$, df $= 8$, $p = 0.00094$), *Acly* ($t = 4.934$, df $= 8$, $p = 0.0011$), *FASN* ($t = 7.186$, df $= 8$, $p = 0.000094$), *ACC2* ($t = 2.697$, df $= 8$, $p = 0.027$)]. Data are presented as mean ± SEM. *$p < 0.05$; **$p < 0.01$; ***$p < 0.001$. n.s. non-significant.

animals daily from P20 to P40 with 2.5 mg/kg of ISRIB (ISR inhibitor). Unphosphorylated eIF2 transfers the methionylated initiator tRNA (Met-tRNA) to the ribosome in a guanosine 5′-triphosphate-dependent manner to start translation[37]. Phosphorylation of the alpha subunit of eIF2 leads to competitive inhibition of eIF2B, the guanosine-exchange factor (GEF) for eIF2, halting translation[38]. ISRIB is a small molecule known to enhance the GEF activity of eIF2B, thereby alleviating the translation block[39,40] (Fig. 7a).

As expected, ISRIB treatment did not change p-eIF2α levels (Fig. 7b), but significantly reduced the upregulation of ATF4 target genes in Phb1-SCKO mice (Fig. 7c). In tibial nerves, we identified a small reduction in myelin thickness (g-ratio = the ratio between axon and fiber diameters) in larger caliber axons of Phb1-SCKO mice, which was accompanied by a reduction in overall axon caliber. However, there was no significant effect of ISRIB on these parameters (Supplementary Fig. 15). Surprisingly, additional morphological analysis of tibial nerves of Phb1-SCKO mice treated with ISRIB revealed a significant exacerbation of the demyelination (Fig. 7d). Supporting this conclusion, quantifications showed an increased number of demyelinated axons and of myelin degradation (myelinophagy) in Phb1-SCKO mice upon ISRIB treatment (Fig. 7e). In line with our morphological findings, Phb1-SCKO animals treated with ISRIB showed a trend toward reduced performance in the rotarod test as compared to Phb1-SCKO mice treated with vehicle (p-values Phb1-SCKO + ISRIB vs Phb1-SCKO + Veh: Day 2 = 0.0936, Day 3 = 0.0979, Day 4 = 0.0723) (Fig. 7f).

Taken together, our results suggest that activation of the ISR is not detrimental and may even be a protective mechanism against demyelination triggered by deletion of Phb1.

## Discussion

In this study, we investigated the role of PHB1 in SCs using conditional knockout mice. We found that ablation of Phb1 from SCs leads to a mild developmental delay in myelination followed by a severe and progressive demyelinating peripheral neuropathy. This contrasts with the persistent myelination arrest that we observed as a consequence of deletion of Phb2 in SCs[8]. Given that PHB1 and PHB2 are thought to primarily act as partners, and that their protein levels are believed to be mutually regulated[23,41], it will be interesting to investigate if this difference reflects a peculiar biology of prohibitins in SCs. In particular, it is possible that PHB1 and PHB2 play diverse roles during developmental myelination and myelin maintenance in the PNS. We found that, when Phb1 is deleted in SCs, the level of PHB2 protein is also reduced. Therefore, we hypothesize that, during development, PHB2 has extra-mitochondrial activities necessary for proper radial sorting, while both PHB1 and PHB2 may be required in mitochondria for long-term myelin maintenance. It is also possible that our results are pointing to previously unappreciated differences in the roles of PHB1 and PHB2, which may also be conserved in other cell types.

Prohibitins can perform a wide range of biological functions and have been described in different cellular compartments[22]. However, both PHB1 and PHB2 are generally concentrated in the mitochondria of all investigated cell types[9]. Accordingly, we report that mitochondria are heavily affected in SCs lacking Phb1, with prohibitins being critical for regulation of mitochondrial morphology, mitochondrial dynamics, mitochondrial membrane potential, mitochondrial respiration, and maintenance of mtDNA. These morphological and functional perturbations to mitochondria accumulate as the demyelinating neuropathy progresses in Phb1-SCKO mice. Unexpectedly, we identified that the pattern of mitochondrial damage progressed to apparent loss of

mitochondria in discrete cells, which correlated with demyelination. Therefore, dysfunction in SC mitochondria seems to be mechanistically linked to demyelination (see schematics in Fig. 7g). This is important because mitochondria with aberrant morphology are often found in SCs from patients affected by neuropathies of several etiologies, including mitochondrial disorders[42], inherited neuropathies[43], and diabetes[44]. Therefore, SC mitochondria are likely important players in peripheral neuropathies.

To our knowledge, there are only a few reports that have explored the impact of the loss of function of mitochondrial proteins in SCs. Conditional deletion of the respiratory chain component Cox10 results in severe dysmyelination (malformed myelin) in the PNS, with many axons remaining unmyelinated despite being correctly sorted[5]. This may suggest that mitochondrial energy production is necessary for proper myelination. However, SCs can tolerate a significant reduction in ATP levels without any noticeable effect in myelin formation or maintenance, as described by our group when Pdha1 (which codes for an essential subunit of the mitochondrial pyruvate dehydrogenase complex) was deleted in SCs[45]. Ablation of the mitochondrial network regulator Gdap1 in SCs leads to an age-related hypomyelinating peripheral neuropathy, recapitulating aspects of the Charcot–Marie–Tooth disease seen in patients with mutated Gdap1[46]. Deletion of mitochondrial m-AAA protease in SCs in adult animals results in the presence of a few demyelinated axons long-term[47]. In addition, Remak SCs lacking m-AAA seem to retract their processes from between the axons[47], an observation similar to our findings in Phb1-SCKO mice at P40. This is interesting since prohibitins are believed to regulate the proteolytic activity of the m-AAA protease[48] and, thus, effects of Phb1 deletion may be partially mediated through the m-AAA protease. Finally, SC-specific ablation of the mitochondrial transcription factor Tfam triggers a progressive demyelinating peripheral neuropathy[6], a phenotype similar to the one observed in Phb1-SCKO mice. Combined, the results from Gdap1, m-AAA, and Tfam conditional knockout mice suggest that SC mitochondrial function is critical for long-term PNS myelin maintenance. Viader et al.[7] proposed that deregulation of lipid metabolism caused by a maladaptive integrated stress response (ISR) could underlie the phenotype seen in Tfam-SCKO mice. We also observed a significant downregulation of enzymes involved with lipid biosynthesis in Phb1-SCKO and, thus, this could be a common perturbation caused by disruption of mitochondrial function in SCs. However, it is still unknown if this alteration in lipid metabolism is solely responsible for the phenotype of Tfam-SCKO mice, and it is possible that there are other mechanisms connecting mitochondrial function to the preservation of peripheral nerves.

A particularly interesting hypothesis is that demyelination happens inadvertently as SCs try to cope with the mitochondrial damage. For this reason, we also investigated how SCs respond to mitochondrial dysfunction. In the absence of Phb1, SCs do not initiate the classical UPR$^{mt}$, first characterized in invertebrates, but instead activate a mitochondrial stress response involving the ISR and ATF4. Indeed, this was recently shown to be the canonical response of mammalian cells to mitochondrial damage[49,50]. Viader et al.[7] proposed that a maladaptive ISR leading to altered lipid metabolism underlies the neuropathy of Tfam-SCKO mice. This hypothesis is particularly attractive since the ISR is known to be causal for nerve pathology in the context of ER stress[51]. With this in mind, we investigated the role of the ISR in Phb1-SCKO mice. Surprisingly, inhibition of the ISR in Phb1-SCKO mice using ISRIB seems to exacerbate the demyelinating neuropathy, indicating that the ISR may be a beneficial response in SCs with perturbed

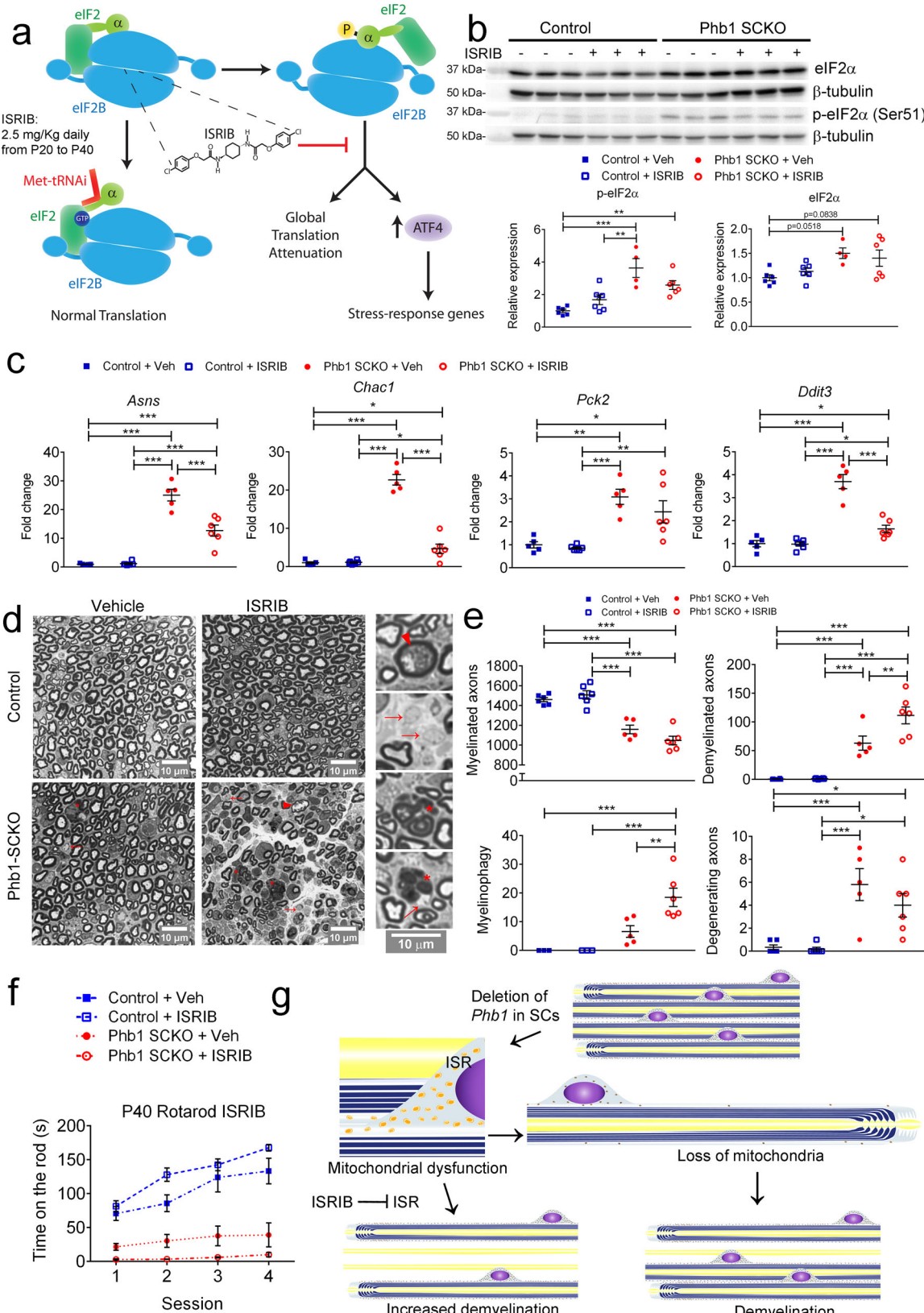

mitochondrial function. An exciting prediction from these findings is that the ISR elicited by mitochondrial dysfunction may have different downstream effects compared to the ISR triggered upon ER stress, something that we would like to explore in the future. It is also possible that, although central to the mitochondrial stress response, the ISR is not the only

pathway activated upon mitochondrial damage. Investigation of these responses in SCs can provide further insight on how mitochondrial dysfunction leads to demyelination.

In conclusion, this study reveals that Phb1-SCKO mice show a phenotype distinct from the previously described Phb2-SCKO animals[8], an unexpected finding that may also be relevant for the

**Fig. 7 Induction of the ISR in Phb1-SCKO mice may be protective against demyelination. a** Schematic of the mechanism of action of ISRIB, an inhibitor of the integrated stress response (ISR). Animals received daily intraperitoneal injections of 2.5 mg/Kg ISRIB or vehicle (Veh) from P20 to P40. **b** ISRIB does not affect eIF2$\alpha$ expression or phosphorylation (western blot from sciatic nerve lysates). $N = 4$-6 animals per group. Two-way ANOVA corrected for multiple comparisons using the Holm-Sidak method. p-eIF2$\alpha$: $F_{(1,18)}$ group $= 32.43$, $p < 0.0001$; $F_{(1,18)}$ interaction $= 7.815$, $p = 0.012$; $p_{Control+Veh\_Phb1SCKO+Veh} = 0.0001$; $p_{Control+Veh\_Phb1SCKO+ISRIB} = 0.005$; $p_{Control+ISRIB\_Phb1SCKO+Veh} = 0.0026$. eIF2$\alpha$: $F_{(1,18)}$ group $= 11.29$; $p < 0.01$. **c** RT-qPCR analyses indicate that the expression levels of several ATF4 genes are reduced upon ISRIB treatment. $N = 5$-6 animals per group. Two-way ANOVA corrected for multiple comparisons using the Holm-Sidak method. *Asns*: $F_{(1,18)}$ group $= 156.4$, $p < 0.001$; $F_{(1,18)}$ treatment $= 18.3$, $p < 0.001$; $F_{(1,18)}$ interaction $= 19.3$, $p < 0.001$; $p_{Control+Veh\_Phb1SCKO+Veh} < 0.0001$; $p_{Control+Veh\_Phb1SCKO+ISRIB} < 0.0001$; $p_{Control+ISRIB\_Phb1SCKO+Veh} < 0.0001$; $p_{Control+ISRIB\_Phb1SCKO+ISRIB} < 0.0001$; $p_{Phb1SCKO+Veh\_Phb1SCKO+ISRIB} < 0.0001$. *Chac1*: $F_{(1,18)}$ group $= 179.7$, $p < 0.001$; $F_{(1,18)}$ treatment $= 90.61$, $p < 0.001$; $F_{(1,18)}$ interaction $= 92.79$, $p < 0.001$; $p_{Control+Veh\_Phb1SCKO+Veh} < 0.0001$; $p_{Control+Veh\_Phb1SCKO+ISRIB} = 0.035$; $p_{Control+ISRIB\_Phb1SCKO+Veh} < 0.0001$; $p_{Control+ISRIB\_Phb1SCKO+ISRIB} = 0.0353$; $p_{Phb1SCKO+Veh\_Phb1SCKO+ISRIB} < 0.0001$. *Pck2*: $F_{(1,18)}$ group $= 34.35$, $p < 0.001$; $p_{Control+Veh\_Phb1SCKO+Veh} = 0.0013$; $p_{Control+Veh\_Phb1SCKO+ISRIB} = 0.013$; $p_{Control+ISRIB\_Phb1SCKO+Veh} = 0.0005$; $p_{Control+ISRIB\_Phb1SCKO+ISRIB} = 0.006$. *Ddit3*: $F_{(1,18)}$ group $= 84.51$, $p < 0.001$; $F_{(1,18)}$ treatment $= 32.15$, $p < 0.001$; $F_{(1,18)}$ interaction $= 31.04$, $p < 0.001$; $p_{Control+Veh\_Phb1SCKO+Veh} < 0.0001$; $p_{Control+Veh\_Phb1SCKO+ISRIB} = 0.045$; $p_{Control+ISRIB\_Phb1SCKO+Veh} < 0.0001$; $p_{Phb1SCKO+Veh\_Phb1SCKO+ISRIB} = 0.045$; $p_{Control+ISRIB\_Phb1SCKO+ISRIB} < 0.0001$. **d** Representative semithin images of tibial nerves. Insets show magnified images. Degenerating axons (arrowhead), demyelinated axons (arrows), myelin degradation (myelinophagy; stars). $N = 5$-6 animals per group. **e** ISRIB treatment leads to increased demyelination and myelinophagy in Phb1-SCKO mice. $N = 5$-6 animals per group. Two-way ANOVA corrected for multiple comparisons using the Holm-Sidak method. Myelinated: $F_{(1,19)}$ group $= 98.01$, $p < 0.001$; $p_{Control+Veh\_Phb1SCKO+Veh} < 0.0001$; $p_{Control+Veh\_Phb1SCKO+ISRIB} < 0.0001$; $p_{Control+ISRIB\_Phb1SCKO+Veh} < 0.0001$; $p_{Control+ISRIB\_Phb1SCKO+ISRIB} < 0.0001$. Demyelinated: $F_{(1,19)}$ group $= 85.58$, $p < 0.001$; $F_{(1,19)}$ treatment $= 6.836$, $p < 0.05$; $F_{(1,19)}$ interaction $= 6.469$, $p < 0.05$; $p_{Control+Veh\_Phb1SCKO+Veh} = 0.0007$; $p_{Control+Veh\_Phb1SCKO+ISRIB} < 0.0001$; $p_{Control+ISRIB\_Phb1SCKO+Veh} = 0.0007$; $p_{Control+ISRIB\_Phb1SCKO+ISRIB} < 0.0001$; $p_{Phb1SCKO+Veh\_Phb1SCKO+ISRIB} = 0.0041$. Myelinophagy: $F_{(1,19)}$ group $= 43.77$, $p < 0.001$; $F_{(1,19)}$ treatment $= 9.938$, $p < 0.01$; $F_{(1,19)}$ interaction $= 9.838$, $< 0.01$; $p_{Control+Veh\_Phb1SCKO+Veh} = 0.078$; $p_{Control+Veh\_Phb1SCKO+ISRIB} < 0.0001$; $p_{Control+ISRIB\_Phb1SCKO+Veh} = 0.078$; $p_{Control+ISRIB\_Phb1SCKO+ISRIB} < 0.0001$; $p_{Phb1SCKO+Veh\_Phb1SCKO+ISRIB} = 0.0014$. Degenerating: $F_{(1,19)}$ group $= 32.17$, $p < 0.001$; $p_{Control+Veh\_Phb1SCKO+Veh} = 0.001$; $p_{Control+Veh\_Phb1SCKO+ISRIB} = 0.013$; $p_{Control+ISRIB\_Phb1SCKO+Veh} = 0.0008$; $p_{Control+ISRIB\_Phb1SCKO+ISRIB} = 0.012$. **f** Phb1-SCKO mice treated with ISRIB were the worst performing group in the rotarod test. Controls are significantly different from Phb1-SCKO mice (irrespectively of treatment; omitted for clarity). $N = 5$-6 animals per group. Repeated measures two-way ANOVA corrected for multiple comparisons using the Holm-Sidak method. $F_{(3,19)}$ group $= 38.06$, $p < 0.001$; $F_{(3,57)}$ time $= 37.24$, $p < 0.001$; $F_{(9,57)}$ interaction $= 7.511$, $p < 0.001$; ($p_{Phb1SCKO+ISRIB\_Phb1-SCKO+Veh}$: Day 2 $= 0.0936$, Day 3 $= 0.0979$, Day 4 $= 0.0723$). **g** Schematic representation of the phenotype seen in Phb1-SCKO mice. Data are presented as mean $\pm$ SEM. $*p < 0.05$, $**p < 0.01$, $***p < 0.001$. Non-significant results omitted for clarity.

study of the biology of prohibitins in other cell types. The rapid, demyelinating peripheral neuropathy observed in Phb1-SCKO mice seems to be significantly more severe than that of previous conditional knockout mice for mitochondrial genes. This may be due to the extensive roles that prohibitins play in mitochondrial biology. One important aspect of mitochondrial function modulated by prohibitins is mitochondrial dynamics. Interestingly, we show that Phb1-SCKO mice seem to have impaired mitochondrial dynamics, accumulation of mitochondrial damage and preferential demyelination of cells with apparent mitochondrial loss. Further studies should investigate the cellular alterations in SCs that link mitochondrial damage to demyelination; but, according to our data, it is unlikely that this mechanism involves the ISR. In fact, it seems that the mitochondrially-induced ISR is beneficial in the context of demyelination in the PNS. Our study adds to the growing body of research demonstrating the crucial role for SC mitochondria to maintain nerve homeostasis. Given the results of this and other reports, perturbations of SC mitochondria are sufficient to elicit severe nerve damage and, therefore, should be viewed as possible mechanisms underlying peripheral neuropathies.

## Methods

**Animal models**. All animal experiments followed ethical regulations for animal testing and research and were approved by the Institutional Animal Care and Use Committee (IACUC) of the Roswell Park Cancer Institute and the regulatory authorities at the University at Buffalo under protocols UB1188M and UB1194M. Animals were housed in individually ventilated cages, separated by gender in groups of at most five per cage, with food and water ad labium, in a room kept at 70 °F ± 2 °F temperature, 30–70% humidity and a 12 h light/dark cycle. Mpz-Cre[13] and Phb1-floxed animals[12] were used in this study. Mice were also crossed to Thy1-YFP[52,53] and PhAM[29] reporter lines. Animals were kept in a C57BL/6 and 129 mixed genetic backgrounds and analyses were only performed from littermates. Animals carrying one or two floxed *Phb1* alleles but no Cre were used as controls, unless otherwise specified. No animals were excluded from this study. Genotyping was performed from genomic DNA as described below. For Mpz-Cre, genotyping primers used were F: 5′ ccaccacctctccattgcac 3′ and R: 5′ gctggcccaaatgttgctgg 3′ and PCR conditions were 94 °C for 5 min, (94 °C for 30 s, 56 °C for 30 s and 72 °C for 1 min) for 30 cycles and 72 °C for

10 min, yielding a ~450 bp band. For PHB1, primers used were P1: 5′ taagactgggtcctgccatt 3′, P2: 5′ gtgcttgcatcagagtcagg 3′ and P3: 5′ ctgtgcccaacaaagcctat 3′ and PCR conditions were 94 °C for 10 min, (94 °C for 40 s, 57 °C for 40 s and 72 °C for 40 s) for 30 cycles and 72 °C for 10 min. Primers P1 and P2 were used for genotyping and P1 and P3 for recombination, yielding bands with 195, 118, and ~260 bp for floxed, WT, and recombined alleles, respectively. For Thy1-YFP, genotyping primers used were F: 5′ acagacacacacccagga 3′ and R: 5′ cggtggtgcagatgaactt 3′ and PCR conditions were 94 °C for 4 min, (94 °C for 20 s, 65 °C for 15 sec and 68 °C for 10 s) in the first 10 cycles, (94 °C for 15 sec, 60 °C for 15 sec and 72 °C for 10 s) for 28 cycles and 78 °C for 5 min. The Thy1-YFP transgene resulted in a 400 bp band. For the PhAM PCR, primers used were F: 5′ ccaaagtcgctctgagttgttatc 3′ WT R: 5′ gagcgggagaaatggatatg 3′ and PhAM R: 5′ caatgggcggggggtcgtt 3′. PCR conditions were 95 °C for 3 min, (95 °C for 30 s, 56 °C for 40 s and 72 °C for 1 min) for 30 cycles and 72 °C for 7 min. The PhAM transgene resulted in a ~350 bp band, while the WT yielded a band at ~650 bp. ISRIB (Cayman chemicals # 16258) was prepared fresh every day by solubilizing ISRIB in a 1:1 mixture of DMSO and PEG400 (Sigma-Aldrich 202398) for a concentration of 2.5 mg/mL. This solution was then heated at 37 °C and vortexed until clear. Mice were administered intraperitoneally with 2.5 mg/kg ISRIB or vehicle daily from P20 to P40.

**Morphological assessments**. Mice were euthanized at the indicated ages and sciatic, tibial, femoral motor, and saphenous nerves were dissected, fixed in 2% glutaraldehyde, and stored at 4 °C until processing. Tissue was then post fixed in 1% osmium tetroxide, dehydrated using sequential incubation in ethanol of increasing concentration, and embedded in Epon resin using propylene oxide as a transition solvent. Semithin sections (1 μm thick) were stained with 2% toluidine blue. Ultrathin sections (80–85 nm) were stained with uranyl acetate and lead citrate to be examined by electron microscopy. For g-ratio analyses, four images per nerve were acquired using the ×100 objective of a Leica DM6000 microscope running the Q-Capture Pro V7.0.4324.5 software (QImaging, Inc.). Axon and fiber diameters were calculated using a semi-automated protocol in the Leica QWin software (Leica Microsystem). For morphological quantifications, images were acquired with the ×100 objective and stitched using the PTGui software v.10 (New House Internet Services BV) to reconstruct a complete image of the nerve. Morphological parameters were then evaluated in the full nerve. Electron micrographs at ×2900 were used for all the quantifications described below. For the mitochondrial perimeter, at least 100 mitochondria per animal were evaluated. For quantifications of axonal size, random images were selected and about 100 myelinated axons and 50 demyelinated axons were measured per animal. For analysis of Remak bundles, about 30 random fields were assessed. Unless stated otherwise, quantifications were performed using ImageJ Fiji v1.52p[54,55].

**Behavioral and electrophysiological analyses**. For rotarod animals were tested in two daily sessions (minimum 6 h rest in between) for two consecutive days on an accelerating rotarod (4–40 rpm in 5 min). Each session included three trials, and the average time on the rod per session was reported. For the electrophysiological analyses, animals were anesthetized using 0.4 mg/g of body weight of 2,2,2-Tri-bromoethanol (Sigma-Aldrich T48402) and maintained under a heat lamp during the procedure. Nerve conduction velocity and amplitude were obtained from sciatic nerves using a Medelec Synergy electromyography device and subdermal steel monopolar needle electrodes. The recording electrode was positioned in the muscle in the middle of the paw, while a reference electrode was positioned in between the digits. Stimulation was performed using a pair of electrodes positioned sequentially at three different points: at the level of the ankle, sciatic notch and in the paraspinal region at the level of the iliac crest. Distal amplitude and average nerve conduction velocity are reported.

**Cell culture and *Phb1* knockdown**. Primary rat SCs were isolated using the Brockes' method[56]. Briefly, P3 rat sciatic nerves were dissected, dissociated using collagenase, and cells were platted on Petri dishes coated with Poly-L-lysine. To kill contaminant fibroblast, cells were treated with Cytosine β-D-arabinofuranoside hydrochloride (Sigma-Aldrich C6645) and two rounds of treatment with anti Thy1.1 antibody (Biorad MCA04G) and rabbit complement. At the end of the purification process, the purity of SCs was 99–100%. Cells were not passaged more than four times and were maintained in media containing high glucose DMEM (4.5 g/L glucose) supplemented with 10% fetal bovine serum (FBS), 2 mM L-glutamine, 100 U/mL penicillin, 100 μg/mL streptomycin, 2 ng/ml Nrg1 (human NRG1-β1 extracellular domain, R&D Systems), and 2 μM forskolin. For the knockdown of *Phb1*, lentivirus was produced in HEK293T cells (NGVB, ngvbcc.org) maintained in DMEM supplemented with 10% FBS, 2 mM L-Glutamine, and 1× MEM Non-Essential Amino Acids (Gibco, 11140-050). Media was changed 2 h prior to transfection. Cells were transfected using a Calcium phosphate, the ViraPower™ Lentiviral Packaging Mix (Thermo Fischer Scientific K497500), and the shRNA constructs (#1 = TRCN088454 and #2 = TRCN087986; Thermo Fisher Scientific). A scramble shRNA construct was used as control. Media was changed 16 h after transfection, and viral particles were allowed to accumulate in the media for 30 h. The supernatant was filtered (0.22 μm), centrifuged at 30,000 × g, resuspended in PBS, and stored at −80 °C until use. Virions were titrated by RT-qPCR using the LV900 (ABM) kit. Primary rat SCs were transduced with 1 million viral particles per 1.9 cm² of growth area and kept in culture for 72 h after transduction. Mouse primary SCs were prepared using a protocol modified from[57]: adult sciatic nerves from 2 to 3 animals per genotype were dissected, stripped from epineurium and other contaminant tissues, and kept in culture for 7 days to allow the formation of repair SCs. Media formulation was identical to media used for rat SCs, except for use of 10 ng/ml Nrg1 (human NRG1-β1 extracellular domain, R&D Systems). After 7 days, cells were dissociated enzymatically using a mixture of 2.5 mg/mL of dispase II (Sigma-Aldrich) and 130 U/mL of type I collagenase (Worthington Biochemical Corporation) for 3 h at 37 °C, mechanically dissociated using fire-polished glass pipettes and seeded on one 35 mm dish coated with laminin (Sigma-Aldrich). Mouse SCs were maintained for up to a week in high glucose DMEM (4.5 g/L glucose) supplemented with 2 mM L-glutamine, 100 U/mL penicillin, 100 μg/mL streptomycin, 2 ng/ml Nrg1 (human NRG1-β1 extracellular domain, R&D Systems), and N2 supplement (Thermo Fisher Scientific). Cells were passaged once onto the appropriate plates for each experiment.

**Seahorse analysis**. Cellular bioenergetics analysis was performed using a Seahorse XFp instrument and the mitochondrial stress test (Agilent), following the manufacturer's instructions. Primary rat SCs were plated on poly-L-lysine coated wells of the XFp plate in a number that resulted in a confluent layer of cells by the time of the analysis: 6000 cells in each well to be transduced with control shRNA and 12,000 cells in each well to be transduced with the *Phb1* shRNAs. The difference in seeding density aimed to equilibrate the cell number at the time of the experiment, since *Phb1* knockdown caused cell death. Transduction with shRNAs was performed 24 h after cell seeding and the Seahorse analysis was carried out 72 h after transduction. For the Seahorse experiment, 1.5 μM oligomycin was used to inhibit mitochondrial ATP synthesis, 2 μM FCCP was used to stimulate maximal mitochondrial respiration, and 0.5 μM rotenone and 0.5 μM antimycin A were used to completely block mitochondrial respiration. During the assay, cells were kept in non-buffered XF DMEM medium pH 7.4 supplemented with 1 mM pyruvate, 2 mM glutamine, and 10 mM glucose. At the end of the assay, cell nuclei were stained with Hoechst 33342 (Thermo Fisher Scientific) and a Biotek Cytation 5 plate reader was utilized to obtain an automated cell count. Parameters of mitochondrial respiration were calculated as follows: basal respiration: (last measurement before oligomycin injection)-(non-mitochondrial oxygen consumption); maximal respiration: (maximum rate measurement after FCCP injection)-(non-mitochondrial oxygen consumption); spare respiratory capacity: (maximum respiration)-(basal respiration). Final results were normalized by cell number.

**Immunofluorescence and quantifications**. For analysis of Thy1-YFP, tibial nerves were dissected, fixed in 4% PFA for 30 min and tissue was whole-mounted for visualization. The general immunofluorescence protocol involved permeabilization

of the tissue using acetone or methanol, blocking for 1 h at room temperature, incubation with primary antibodies overnight at 4 °C, incubation with secondary antibodies for 1 h at room temperature, counterstaining with DAPI, and mounting of slides with Vectashield (Vector Laboratories). Blocking buffer 1 was used for sections and contained 20% FBS, 1% bovine serum albumin and 0.1% Triton X-100 in 1X PBS. Blocking buffer 2 was used for teased fibers and contained 5% fish skin gelatin and 0.1% Triton X-100 in 1× PBS. Below we describe the particularities of each experiment. For analyses involving the PhAM transgene, sciatic nerves were dissected, fixed in 4% PFA for 30 min, washed with PBS and teased in slides coated with (3-aminopropyl) triethoxysilane (TESPA; Sigma-Aldrich). Coating with TESPA was achieved by subsequently submerging glass slides in acetone for 1 min, 4% TESPA in acetone for 2 min and two times in acetone for 30 s each. The teasing procedure consisted in placing a small portion of the nerve in a PBS droplet over the TESPA-coated slide, followed by careful mechanical separation of individual fibers, first using insulin syringes (0.3 ml 31 G × 8 mm) and then using modified insulin syringes containing insect pins (Fine science tools #26002-10) attached to their needle. Slides were allowed to dry for at least 1 h before staining. For co-staining, the following primary antibodies were used: rabbit anti-KDEL 1:500 (Thermo Fisher Scientific #PA1-013), rabbit anti-Kv1.1 1:200 (Alomone #APC-009), rabbit anti-p75[NTR] 1:1500 (Cell signaling #8238), rabbit anti-S100β 1:200 (Dako #ZO311), chicken anti-Neurofascin 1:1000 (R&D Systems #AF3235), rabbit anti-TOM20 1:500 (Proteintech #11802-1-AP), rabbit anti-HSPD1 1:500 (Proteintech #15282-1-AP), chicken anti-P0 1:300 (Aves #PZO0308) and Mouse anti-MBP (Smi99) 1:500 (Biolegend #808401). Staining for TOM20 required no permeabilization. For quantification of mitochondrial size and location, 1024 × 1024 pixel z-stacks were acquired with a ×100 objective and ×1.5 zoom, with a 0.13 μm z-step size to allow oversampling of the z plane. The voxel size was 103.3 μm × 103.3 μm × 125.9 μm (width × height × depth). Quantifications were performed automatically using the 3D ImageJ Suite[58] on ImageJ Fiji v1.52p. The code for the automated routine is available at https://github.com/gdfnunes/3D-analysis-mitochondria. Mitochondria were considered in the vicinity of a given cellular location if they were no more than 2 μm distant from that location. Secondary antibodies used in this study were Alexa 488 donkey anti-rabbit IgG 1/1000 (Jackson ImmunoResearch #711-545-152), Cy3 donkey anti-chicken IgY 1/500 (Jackson ImmunoResearch #703-165-155), Cy3 donkey anti-mouse IgG 1/500 (Jackson ImmunoResearch #715-165-150), and rhodamine (TRITC) anti-rabbit IgG 1/500 (Jackson ImmunoResearch #711-025-152). Images for all the above experiments were acquired using a confocal microscope Leica SP5II running the LAS AF 2.7.9723.3 software (Leica).

**Live imaging**. In all live imaging experiments, SCs were maintained in media prepared with Fluorobrite DMEM (Thermo Fischer Scientific A1896701) and with N2 supplement (Thermo Fisher Scientific) instead of regular DMEM and FBS. For analyses of mitochondrial dynamics and mitophagy, cells were maintained at 37 °C and 5% CO₂ using a live imaging stage coupled to the Leica SP5II confocal microscope. For assessment of mitochondrial dynamics, primary mouse SCs were isolated from control and Phb1-SCKO mouse expressing mito-dendra2 (PhAM reporter) and plated in Lab-Tek chamber slides coated with laminin. In each experiment, we pooled cells from 2 to 3 animals for each genotype. Data reported represent results from two independent experiments. For each cell, a circular ROI of the same size was stablished about a third of the way in one of its processes. To promote photoconversion of mito-dendra2, this region was stimulated with the 405 nm laser at 20% power for 128 iterations. Cells were then monitored for 30 min using the 488 nm and the 561 nm laser lines to stimulate mito-dendra2 in the unconverted and photoconverted state, respectively. Images were obtained using a ×40 objective and ×3 zoom. Post-acquisition, intensity on the red channel was analyzed using the Dynamic ROI Profiler on ImageJ Fiji v1.52p by stablishing a line with its center passing through the photoconverted area. For analysis of mitochondrial membrane potential, primary mouse SCs were isolated from P40 control and Phb1-SCKO mice (pool of four animals each). 10,000 SCs were seeded in a glass-bottom 96-well plate coated with laminin. Cells were incubated for 30 min with 50 nM Mitotracker green (Thermo Fischer Scientific # M7514) and 20 nM TMRM (Thermo Fischer Scientific # I34361) and immediately imaged using a Biotek Cytation 5 plate reader. Hepes 25 nM was added to maintain the pH and plates were kept at 37 °C during incubation and acquisition. 2 μM FCCP was used as a positive control to cause mitochondrial depolarization. Four images per well were acquired from six wells per genotype. Cells with morphology not compatible with SCs (Fig. 4d) were excluded from the analysis. Quantifications were performed using auto-thresholding in ImageJ Fiji v1.52p and the ratio between the TMRM and Mitotracker green signals is reported.

**Western blot**. Sciatic nerves were dissected, epineurium and other contaminant tissues were removed and samples were snap frozen in liquid nitrogen and stored at −80 °C until processing (except for the analysis of p-PERK, in which protein extraction and SDS-PAGE were carried out immediately after dissection). Nerves were pulverized and resuspended in lysis buffer [50 mM Tris pH 7.4, 150 mM NaCl, 1% IGEPAL CA-630, 1 mM EDTA, 1 mM EGTA, 0.1% SDS, 0.5% sodium deoxycholate, 1 mM sodium orthovanadate, 1 mM sodium fluoride, protease inhibitor cocktail (Sigma-Aldrich P8340), phosphatase inhibitor cocktail 2 (Sigma-Aldrich P5726) and phosphatase inhibitor cocktail 3 (Sigma-Aldrich P0044)]. After lysis, samples were sonicated in a water sonicator with three cycles of 20 s at

70% power and then centrifuged at $13,200 \times g$ for 15 min at 4 °C. The supernatant was collected and protein concentration was determined by BCA protein assay (Thermo Scientific) according to the manufacturer's instructions. Equal amounts of protein per sample were diluted 3:1 in 4× Laemmli (250 mM Tris-HCl pH 6.8, 8% SDS, 5% β-Mercaptoethanol, 40% Glycerol, 0.04% Bromophenol blue) and denatured for 5 min at 100 °C. Samples were then loaded on an SDS-polyacrylamide gel and electroblotted onto a PVDF membrane. Blots were then blocked with 5% BSA in TBS-T (1× TBS + 0.1% Tween-20) and incubated overnight with the appropriate primary antibody: PHB1 1:500 (Abcam #ab28172), PHB2 1:250 (Millipore # ab10198), Opa1 1:500 (BD Biosciences # 612606), Erk1/2 1:500 (Cell signaling # 9102), p-ERK1/2 1:500 (Cell signaling # 9101), β-tubulin 1:5000 (Novus Biologicals #NB600-936), TOM20 1:500 (BD Biosciences # 612278), GAPDH 1:5000 (Sigma-Aldrich #G9545), eIF2α 1:500 (Cell signaling #5324), p-eIF2α 1:500 (Cell signaling #3398), Clpp 1:500 (Proteintech #15698-1-AP), Hspd1 1:500 (Proteintech # 15282-1-AP), PERK 1:500 (Cell signaling #3192), p-PERK 1:500 (Cell signaling #3179), Bip 1:500 (Novus Biologicals #NB300-520), ACC 1:500 (Cell signaling # 3662), p-ACC 1:500 (Cell signaling #3661). Membranes were then rinsed in 1× TBS-T and incubated for 1 h with secondary antibodies. Blots were either imaged directly with Odyssey CLx infrared imaging system (Li-Cor) or developed using ECL Select (GE Healthcare) and imaged using a ChemiDoc XRS system. Quantifications were carried out in the Image lab 6.0 software (Biorad) for blots imaged with the ChemiDoc XRS or in the Image Studio Lite 5.2 (Odyssey) for blots imaged with the Odyssey CLx. GAPDH or β-tubulin were used as the loading control. All uncropped blots are presented in Supplementary Figs. 16 and 17.

**RNA extraction and RT-qPCR analyses.** Sciatic nerves were dissected, stripped from epineurium, flash frozen in liquid nitrogen, and stored at −80 °C until processing. Total RNA was isolated using Trizol (Thermo Fisher Scientific) and reverse transcribed using the Superscript III kit (Thermo Fisher Scientific #18080051). For each reaction, 1 µg of RNA, 5 µM of oligo(dT)20, and 5 ng/µl random hexamers were used. RT-qPCR for lipid metabolism and mitochondrial transcripts used the SYBR green method. qPCR using the SYBR green method was also used for the quantification of mtDNA/gDNA, which were extracted from sciatic nerves using Phenol/Chloroform. For quantification of spliced Xbp1 (sXbp1), we used the Taqman assay-on-Demand Mm03464496_m1 (Applied Biosystems) and normalized data to results using probe Mm99999915_g1 to detect the reference gene *Gapdh*. *Gapdh* analyzed through the Taqman method was also utilized to normalize the expression of prohibitins over time. All of the other assays were carried out using the Universal Probe library from Roche Diagnostic and the Faststart Universal Probe Master (Rox) (Sigma-Aldrich #4913949001) as suggested by the manufacturer. UPL and SYBR data were normalized to the reference gene β-actin. All assays were performed using a Bio-Rad CFX96/384 real-time PCR machine using the following cycle: SYBR and Taqman: 95 °C for 10 min and (95 °C for 15 s and 60 °C for 1 min) for 40 cycles; UPL: 50 °C for 2 min, 95 °C for 10 min and (95 °C for 10 sec and 60 °C for 30 s) for 40 cycles. Data were analyzed using the threshold cycle (Ct) and 2(−ΔΔCt) methods, and the average expression of control animals was normalized to 1. A full list of primers used in this study is available in Supplementary Table 1.

**Statistical analyses.** Experiments (with the exception of assigning animals to ISRIB or Veh groups) were not randomized, but all data collection and analysis were performed blind to the conditions of the experiments and genotype of the mice where applicable. No data were excluded from the analyses. No power analysis was performed, but our sample sizes are similar to those generally used in the field. The statistical test used in each analysis is reported in the legend of every figure. Data are presented as mean ± SEM. Values of $P < 0.05$ were considered to represent a significant difference, while $0.05 < P < 0.1$ was considered to represent a trend. Data were analyzed using GraphPad Prism 6.01.

**Reporting summary.** Further information on research design is available in the Nature Research Reporting Summary linked to this article.

## Data availability
The data supporting the findings of this study are available within the article and its supplementary information files. All original data and biological resources (mouse strains, plasmids, etc.) are available from the corresponding author upon reasonable request. Source data are provided with this paper.

## Code availability
The code of the Image J macro used to quantify mitochondrial size can be found at https://github.com/gdfnunes/3D-analysis-mitochondria (https://doi.org/10.5281/zenodo.4671062)[59].

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

## Acknowledgements

We thank J. Catlin and S. S. Rasam for assistance in performing the UPRmt western blot and lipid peroxidation (TBARS) experiments, respectively. We thank M. R. Weaver for the careful English Language revision. This work was funded by grant NIH-NINDS-R01NS100464 (to M.L.F.). Generation of the Phb1-floxed animals was originally supported by National Institutes of Health Grants HD08818 and HD07857 to B.W.O.

## Author contributions

G.D.N., Y.P., and M.L.F. designed research and interpreted data; G.D.N. Carried out the majority of the experiments; Y.P. performed the proliferation and TUNEL analyses in 20-day-old mice; L.N.M. and E.H. prepared and processed the tissue for morphological analyses; N.S. performed the electrophysiology analysis; B.H. and B.W.O. provided the Phb1-floxed mice; B.B. provided technical assistance for the Seahorse analysis; G.D.N. and M.L.F. wrote the manuscript; E.R.W., Y.P., L.W., B.B., B.W.O., and B.H. analyzed the data and critically reviewed the manuscript.

## Competing interests

The authors declare no competing interests.
