## [Peer Review File · Nature Communications]

REVIEWER COMMENTS

Reviewer #1 (Remarks to the Author):

The study reports that deletion of the prohibitin 1 gene (Phb1) in SCs causes mitochondrial damage and demyelinating peripheral neuropathy. In contrast to deletion of Phb2 which causes developmental changes in myelination, Phb1 appears to be required for maintaining myelin. The conditional SC phb1 knockouts show signs of ISR and decreases in lipid synthesis enzymes. This neuropathy seems to occur in both sensory and motor fibers. They examined different time points, P20, P40, P90 to determine the sequence of events. They rule out macrophage infiltration and erk activation as causes of the demyelination. Their data point to a link between mitochondrial damage and demyelination. These findings add to the growing understanding of the importance of mitochondria to myelination.

In general the study is well designed and presented, however their conclusions would be strengthened with some clarification.

1. In Fig.3A a decrease in mitochondrial DNA is shown. If this is the case, are there decreases in mitochondrial encoded transcripts?

2. In Fig 3F,G,H respirometry was performed in primary SCs transfected with shRNA to phb1. In the respirometry trace it would be helpful to show where oligomycin, FCCP, rotenone/AA were added, but most importantly, these cells don't appear to have any spare respiratory capacity. Also, the oxygen consumption rates (pmol/min) are very low. The spare respiratory capacity should be calculated by subtracting basal respiration from maximal respiration after FCCP, after first subtracting out nonmitochondrial respiration from both basal and maximal. This data isn't clear. In Fig 3F basal is as high or higher than maximal, which would indicate that there is no spare respiratory capacity?

3. They also show decreased Opa1, or a degradation of Opa 1, indicating that fusion should be impaired, however most mitochondria appeared larger. This should be clarified.

4. In Fig 3I the efficiency of the shRNA knockdown for their two shRNAs of phb1 is 63% and 21%. Why do you think there is really no difference in effects on respiration?

Minor points:

The paper is well written, but very dense. To reach a wider audience it would be helpful to explain some of the approaches and structures. For example, in Fig 3B you show that with the phb1 knockout mice crossed with the PhAM reporter mice, that the red signal remains after photoconversion. In the results it's stated that this indicates a change in mitochondrial dynamics or transport. A sentence on fusion and fission and what the photoconversion is doing might help. Also for Fig. 2 it would help to show a schematic of an axon showing the nodes, paranodes.

In lines 136, 137, "the distribution of demyelinated axons", are you referring to axon caliber here? This isn't clear.

In Supplementary Fig. 1 the levels of phb1 protein are reduced in the phb1 knockouts. This is in total

lysates, is there a change in mitochondrial fractions?

Reviewer #2 (Remarks to the Author):

The manuscript builds on extensive previous work of the Feltri lab on peripheral nerve (and, in particular, Schwann cell) biology. This is an important topic with regard to understanding peripheral neuropathy pathogenesis, and may potentially also contribute to the development of nerve injury treatments.

Recently, the Feltri group has focused on the role of prohibitins (Phb) in peripheral nerve myelination. Phb1 and 2, which are highly homologous to each other, exist in large ring-like multimeric complexes composed of heterodimers, and serve pleiotropic (most prominently, mitochondrial) functions. Being involved in regulating OPA1 isoform expression at the mitochondrial inner membrane, and thereby regulating essential functions of these organelles, prohibitins are essential for mitochondrial fusion. In mice, their deletion is embryonically lethal.

The authors now show that Schwann cell (SC)-specific Phb1 depletion leads to diverse mitochondrial perturbations (including mtDNA loss), changes in lipid biosynthesis, and an activation of the integrated stress response (ISR). In Phb1-SCKO mice, this leads to a demyelinating neuropathy (culminating in a motor-phenotype), which worsens upon pharmacological inhibition of the ISR, leading the authors to conclude that in their model, ISR activation is beneficial. These are novel and important findings.

General:

The current study uses a comprehensive repertoire of experimental protocols. It is thoroughly conducted, and mechanistically well-developed. In my eyes, the paper's significance could be considerably enhanced by a demonstration of whether the peripheral demyelinating neuropathy/motor phenotype of Phb1-SCKO mice can be rescued upon reversing the knockout. Will restoring mitochondrial function lead to remyelination?

Specific:

I only have the following minor comments:

1. To improve readability, I'd advise to introduce the non-nerve specialists among the readership to the following terms: Remak bundle, Cajal bands, juxtanode, paranode.
2. The manuscript would significantly benefit from proof-reading /editing by a native speaker.
3. Suppl. Figs 1A, B: As PHB1 is thought to exist in multimeric complexes composed of heterodimers with PHB2: are PHB2 levels affected upon Phb1 KO? If yes, the authors should generally be cautious throughout their manuscript attributing observed effects to the deletion of Phb1, as PHB2 is likely degraded in the absence of its assembly partner (reviewed by Merkwirth C & Langer T, BBA, 2009) .
4. Suppl. Fig 4: which nerve was examined here?
5. Suppl. Fig 6: phospho-specific antibodies are notoriously problematic with regard to specificity, and this also applies to anti-pHiston 3 (which the authors used to quantify mitotic activity). Here,

staining for a proliferation-associated antigen such as Ki67 is perhaps much more informative (and technically much more robust).

6. Suppl. Fig 8: not clear, which timepoint (P20?, P40?) is shown.

7. There are many typos, just e.g. line 104 : ablation of Phb1 does not impairs, line 176: phagocytose; line 348: ISRIB; Figure 6G: mitochondrial dysfunction; Suppl Fig 7 and 9 (legend): Gaussian, and so on and so on (see my specific comment 2).

Reviewer #3 (Remarks to the Author):

The manuscript by Della-Flora Nunes et al. represents a strong addition to the understanding of peripheral myelin maintenance by describing the important function of mitochondrial prohibitin1 in Schwann cell metabolism. The authors performed a comprehensive investigation *in vivo*, combining a wide array of elegant techniques and analyses. This manuscript will likely be seen as a major contribution to the ever-improving understanding of peripheral nervous system myelination and demyelinating peripheral neuropathies.

Previously, the authors identified novel molecules involved in axo-glial interaction and peripheral myelination, including members of the prohibitin family, and described the role of prohibitin 2 in Schwann cell radial sorting and myelination *in vivo*.

In the present study, Della-Flora Nunes et al. found that unlike prohibitin 2, prohibitin 1 plays little to no role in developmental myelination, but is critical for peripheral myelin maintenance. This is a very interesting finding, however, I am concerned that because of the lack of expression data in the manuscript, the authors cannot rule out the hypothesis that prohibitin 1 and prohibitin 2 expression patterns are distinct, and that prohibitin 1 might be predominantly expressed in myelinating Schwann cells, and not at developmental stages (whereas prohibitin 2 might be expressed earlier over the course of Schwann cell development). I think it is worth investigating the transcript or protein levels of both paralogs (Phb1 and Phb2) at a timepoint corresponding to developmental myelination in WT animals, and at P20, which corresponds to the beginning of the Phb1 peripheral neuropathy. This would bring the missing piece needed to fully support the conclusion of this study. Another hypothesis is that a compensatory mechanism results in an increase in prohibitin 2 transcripts in Phb1-SCKO mice at least prior to and at the onset of myelination. In section 2.4 of the results, because prohibitins are required for the Raf-MEK-ERK cascade, the authors tested the expression and phosphorylation of ERK1/2 in Phb1-SCKO mice. As reported in Supplementary Figures 5A and B, they found little change in total and/or phosphorylated-ERK. Again, this could be explained by an upregulation of prohibitin 2, but I understand that investigating potential genetic compensatory mechanisms is beyond the scope of this paper.

The following is not a concern, but more out of curiosity. When analyzing the Remak bundles, the authors noted the presence of Remak SCs retracting their processes and they hypothesized that they might undergo transdifferentiation to promote nerve repair. Did the authors observe elevated c-Jun in these Remak SCs?

One minor detail I would add relates to the graphs. The quantifications and graphs are all rigorous and convincing. Adding n.s. (or equivalent mention) when the comparisons are not statistically

significant would help the reader sort the important message out of the data, which is quite considerable.

RESPONSE TO REVIEWER COMMENTS

Reviewer #1 (Remarks to the Author):

The study reports that deletion the prohibitin 1 gene (Phb1) in SCs causes mitochondrial damage and demyelinating peripheral neuropathy. In contrast to deletion of Phb2 which causes developmental changes in myelination, Phb1 appears to be required for maintaining myelin. The conditional SC phb1 knockouts show signs of ISR and decreases in lipid synthesis enzymes. This neuropathy seems to occur in both sensory and motor fibers. They examined different time points, P20, P40, P90 to determine the sequence of events. They rule out macrophage infiltration and erk activation as causes of the demyelination. Their data point to a link between mitochondrial damage and demyelination. These findings add to the growing understanding of the importance of mitochondria to myelination. In general the study is well designed and presented, however their conclusions would be strengthened with some clarification.

We thank the reviewer for the careful revision and the positive feedback.

1. In Fig.3A a decrease in mitochondrial DNA is shown. If this is the case, are there decreases in mitochondrial encoded transcripts?

We now performed analyses of transcript levels of mitochondrially encoded cytochrome c oxidase I (mt-Co1), mitochondrially encoded NADH dehydrogenase 1 (Nd1) and mitochondrially encoded ATP synthase 6 (mt-Atp6). We found a significant reduction in mRNA level of these transcripts on P40 Phb1-SCKO sciatic nerves in comparison to controls, but no changes at P20, demonstrating the progressive nature of the mitochondrial dysfunction. Similarly, we also found reduced mRNA levels of Tfam and PGC1 α at P40, important transcriptional regulators for mitochondrial function that are encoded in the genomic DNA. On the other hand, we did not identify any changes in transcript levels of Succinate Dehydrogenase Complex Iron Sulfur Subunit B (*Sdhb*), which codes for a component of Complex II, the only mitochondrial complex encoded exclusively from the genomic DNA. We have included these new data in our new Supplementary Figure 10

2. In Fig 3F,G,H respirometry was performed in primary SCs transfected with shRNA to phb1. In the respirometry trace it would be helpful to show where oligomycin, FCCP, rotenone/AA were added, but most importantly, these cells don't appear to have any spare respiratory capacity. Also, the oxygen consumption rates (pmol/min) are very low. The spare respiratory capacity should be calculated by subtracting basal respiration from maximal respiration after FCCP, after first subtracting out nonmitochondrial respiration from both basal and maximal. This data isn't clear. In Fig 3F basal is as high or higher than maximal, which would indicate that there is no spare respiratory capacity?

Thank you for the feedback. We have now signaled in the graph the timing of addition of Oligomycin, FCCP and rotenone/AA. It is true that, in these conditions, SCs transduced with sh-RNAs have a small spare respiratory capacity. We believe that the lentiviral transduction may negatively affect the overall maximum SC respiration since we obtain higher values on experiments with non-transduced SCs. However, even in these conditions, it seems that knockdown of Phb1 has a specific effect, completely ablating the cell's spare respiratory capacity. The spare respiratory capacity was calculated following the instructions by Agilent Seahorse, as explained below:

Non-mitochondrial oxygen consumption: Minimum rate after injection of Rotenone/antimycin A
Basal respiration: (last measurement before Oligomycin injection) – (Non-mitochondrial oxygen consumption)

Maximal respiration: (Maximum rate measurement after FCCP injection) – (Non-mitochondrial oxygen consumption)

Spare Respiratory Capacity: (Maximum respiration) – (Basal respiration)

Therefore, SCs transduced with shRNA targeting *Phb1* have no spare respiratory capacity, while SCs transduced with control shRNA have a spare respiratory capacity of ~1pmol/min/1000 cells.

We have added the description of these calculations to the “Materials and methods” section.

We do not think that the reported oxygen consumption rates are low. Our data has been normalized to 1,000 cells at the end of the experiment, and this is the reason why the OCR seems small. Normalization by cell number is essential for correct interpretation of results and to allow replication of Seahorse studies by other laboratories (see <https://www.agilent.com/cs/library/applications/application-normalization-technical-guidelines-cell-analysis-5994-0022en-agilent.pdf.pdf>). Our data is very similar to other studies in the literature, which report basal OCRs of ~0.5 to ~5.5 pmol/min/1,000 cells in a variety of cell types (Figure 8b on Zhang J. et al., Nat Protoc. 2012 May 10;7(6):1068-85)¹ and basal OCRs of ~2.5 to ~4.5 pmol/min/1,000 cells in SCs (Figure 4i on Pooya S. et al., Nat Commun. 2014 Sep 26;5:4993)².

3. They also show decreased Opa1, or a degradation of Opa 1, indicating that fusion should be impaired, however most mitochondria appeared larger. This should be clarified.

Thank you! We believe that our data reveals two different events that affect mitochondrial morphology in SCs: mitochondrial swelling and defects in mitochondrial dynamics. Due to the anatomical configuration of the nerve, most SC mitochondria are aligned with their longer axis parallel to the direction of the nerve. Therefore, in electron micrographs, mitochondria are cut in cross sections, and the increased perimeter of mitochondria in SCs of *Phb1*-SCKO mice likely reflect the mitochondrial swelling, a pathological finding that can be associated to early compromised mitochondrial function³. At P20, the mitochondrial swelling may be the predominant mitochondrial defect of *Phb1*-SCKO mice, and, consequently, evaluation of mitochondrial volume using the PhAM reporter revealed mostly a slight mitochondria enlargement. On the other hand, as pathology progresses and defects accumulate in the mitochondria, the changes in mitochondrial dynamics become more relevant. As a consequence, we show a relevant loss of mitochondria in P40 *Phb1*-SCKO mice and a fragmentation of the remaining mitochondria Cajal bands of SCs of these animals. One reason why increased proteolytic processing of Opa1 in *Phb1*-SCKO mice may not have more stark consequences is that SC mitochondria seem to be mostly individualized instead of forming longer tubules. Therefore, mitochondrial fragmentation is less evident. We rewrote part of the result section to make this clearer.

4. In Fig 3I the efficiency of the shRNA knockdown for their two shRNAs of *phb1* is 63% and 21%. Why do you think there is really no difference in effects on respiration?

We are not sure if we understood your question correctly.

- If you are asking about the lack of effect of *Phb1* knockdown on the basal respiration: The *in vitro* experiments are very short-term because of the lethality of *Phb1* knockdown in SCs in culture. Therefore, these cells may not have enough time to display the full spectrum of phenotypes caused by loss of *Phb1*. The spare respiratory capacity can capture more subtle changes in mitochondrial activity, and we believe that this is the reason why it is changed first. The consequences of this acute loss of *Phb1* *in vitro* may differ significantly from the long-term loss of *Phb1* *in vivo*. For example, *in vivo* we do not identify a significant level of cell death. On

the other hand, *Phb1* knockdown *in vitro* does not seem to activate the ISR or to cause mitochondrial loss (unpublished data). Therefore, we believe that mitochondrial respiration is much more affected in *Phb1*-SCKO mice, but, due to technical limitations, it is difficult to probe it directly *in vivo*.

- If you are asking why the two different shRNAs result in the same reduction in spare respiratory capacity even though they have different efficiencies:

Indeed, this was also a bit surprising to us. We interpret this as the fact that a small reduction in levels of *Phb1* is already enough to affect the spare respiratory capacity of SCs, at least in culture in these experimental conditions. It seems that the spare respiratory capacity is already completely ablated by the less efficient sh-*Phb1*#2 and, therefore, it cannot be reduced further by the additional efficiency provided by sh-*Phb1*#1. Overall, in different experiments (in this paper and in Poitelon et al. 2015 ⁴), both shRNAs are similarly effective with a slightly better response of sh-*Phb* #1 (Thermo TRCN088454) compared to sh-*Phb* #2 (Thermo TRCN087986).

Minor points:

The paper is well written, but very dense. To reach a wider audience it would be helpful to explain some of the approaches and structures. For example, in Fig 3B you show that with the *phb1* knockout mice crossed with the PhAM reporter mice, that the red signal remains after photoconversion. In the results it's stated that this indicates a change in mitochondrial dynamics or transport. A sentence on fusion and fission and what the photoconversion is doing might help.

Also for Fig. 2 it would help to show a schematic of an axon showing the nodes, paranodes.

Thank you for the suggestions. We have added a description of mitochondrial dynamics and of the photoconversion process to the text. We have also included a schematic representation of the domain structure of a myelinating SC to Figure 3 with a description of important regions, such as the node and paranode.

In lines 136, 137, "the distribution of demyelinated axons", are you referring to axon caliber here? This isn't clear.

Yes, we are. We adjusted this sentence to "...demyelinated axons showed reduced axon caliber compared to myelinated axons in *Phb1*-SCKO animals"

In Supplementary Fig. 1 the levels of *phb1* protein are reduced in the *phb1* knockouts. This is in total lysates, is there a change in mitochondrial fractions?

Yes, we believe that the reduction in total lysates is reflected also by a reduction in the mitochondrial fraction. Indeed, we obtained a similar result in samples enriched for the mitochondrial fraction. The mitochondrial fraction was prepared with a protocol adapted from Viader et al., 2011 ⁵. Briefly, four P20 sciatic nerves per sample were dissected, desheathed, mechanically dissociated and digested in 10% collagenase for 20 min at 37C. The sample was then centrifuged at 500g for 5 min, and the pellet was homogenized in 1.5ml of Extraction Buffer (50 mM HEPES, pH 7.5, containing 1 M mannitol, 350 mM sucrose, and 5 mM EGTA, 2 mg/mL BSA) using a glass homogenizer. The homogenate was then subjected to two sequential centrifugations of 10 min at 800g and 8000g, with the supernatant being used in each of the sequential steps. Next, the pellet was resuspended in Extraction Buffer without BSA and the centrifugation steps were repeated. At the end, the pellet was resuspended in 40 μ l of lysis buffer and processed for western blot as described in the manuscript.

With this protocol we obtained a ~45% enrichment on *Phb1* levels in the mitochondrial enriched fraction. We also obtained a significant reduction of proteins in other cellular compartments: cytosol

(GAPDH), nucleus (H2B) and ER (calnexin). However, unfortunately, this protocol resulted in myelin (P0) as a major contaminant. For this reason, and because of the similarity of the results to the experiment on whole sciatic nerve lysates, we opted not to include this data in the paper. The result for P20 animals is illustrated in **Supporting Figure 1** below.

Supporting Figure 1. (a) Representative WB of SN lysates of P20 mice before or after mitochondrial enrichment. (b) Quantification of PHB1 levels. # non-specific band. * $P < 0.05$.

Reviewer #2 (Remarks to the Author):

The manuscript builds on extensive previous work of the Feltri lab on peripheral nerve (and, in particular, Schwann cell) biology. This is an important topic with regard to understanding peripheral neuropathy pathogenesis, and may potentially also contribute to the development of nerve injury treatments.

Recently, the Feltri group has focused on the role of prohibitins (Phb) in peripheral nerve myelination. Phb1 and 2, which are highly homologous to each other, exist in large ring-like multimeric complexes composed of heterodimers, and serve pleiotropic (most prominently, mitochondrial) functions. Being involved in regulating OPA1 isoform expression at the mitochondrial inner membrane, and thereby regulating essential functions of these organelles, prohibitins are essential for mitochondrial fusion. In mice, their deletion is embryonically lethal.

The authors now show that Schwann cell (SC)-specific Phb1 depletion leads to diverse mitochondrial perturbations (including mtDNA loss), changes in lipid biosynthesis, and an activation of the integrated stress response (ISR). In Phb1-SCKO mice, this leads to a demyelinating neuropathy (culminating in a motor-phenotype), which worsens upon pharmacological inhibition of the ISR, leading the authors to conclude that in their model, ISR activation is beneficial. These are novel and important findings.

Thank you for all your comments and suggestions and for the thorough revision of the manuscript.

General:

The current study uses a comprehensive repertoire of experimental protocols. It is thoroughly conducted, and mechanistically well-developed. In my eyes, the paper's significance could be considerably enhanced by a demonstration of whether the peripheral demyelinating neuropathy/motor phenotype of Phb1-SCKO mice can be rescued upon reversing the knockout. Will restoring mitochondrial function lead to remyelination?

Thank you for the suggestion! We agree that this is an interesting question. If demyelination could be stopped/rescued in Phb1-SCKO by reestablishing mitochondrial function, the neuropathy could be more decisively ascribed to the mitochondrial defects in Phb1-SCKO mice. Unfortunately, this is a technically challenging experiment and we currently do not have any methodology that would allow reexpression of Phb1 specifically in SCs of Phb1-SCKO mice in a reasonable time frame. This would require either a transgenic mouse that expresses PHB1 specifically in SCs, which does not exist to our knowledge, or to set up injection of viral vectors to infect enough Schwann cells *in vivo*, which is usually low specificity and low efficiency and it would be extremely time consuming. Therefore, we have made attempts to rescue mitochondrial function without necessarily reexpressing Phb1 in SCs. We treated Phb1-SCKO mice with 2.5 mg/kg Mdivi-1 daily from P20 to P40. Mdivi-1 is a putative inhibitor of the mitochondrial fission protein Drp1⁶. By treating Phb1-SCKO mice with Mdivi-1 we were hoping to balance out potential defects in mitochondrial dynamics caused by deletion of Phb1 and consequent proteolytic processing of Opa1. Mdivi-1 had already been demonstrated to have beneficial effects *in vivo* in conditions like diabetes⁷ and kidney injury⁸. However, Mdivi-1 was unfortunately unable to prevent the mitochondrial loss seen in Phb1-SCKO mice and, as a consequence, had no beneficial effect on motor behavior of these animals (**Supporting Figure 2**).

Supporting Figure 2. (a) Representative IHC in teased fibers of P40 Phb1-SCKO mice treated with Mdivi-1 or Vehicle (Veh) daily from P20 to P40. (b) Quantification of mitochondrial loss in each of the groups at P40. (c) Mdivi-1 does not rescue motor deficits of Phb1-SCKO mice.

Specific:

I only have the following minor comments:

1. To improve readability, I'd advise to introduce the non-nerve specialists among the readership to the following terms: Remak bundle, Cajal bands, juxtanode, paranode.

Thank you very much for the suggestion! We now describe all these terms in the text.

2. The manuscript would significantly benefit from proof-reading /editing by a native speaker.

Thank you for your comment. We thoroughly revised our manuscript and sent it for proof-reading by an additional English native speaker.

3. Suppl. Figs 1A, B: As PHB1 is thought to exist in multimeric complexes composed of heterodimers with PHB2: are PHB2 levels affected upon Phb1 KO? If yes, the authors should generally be cautious throughout their manuscript attributing observed effects to the deletion of Phb1, as PHB2 is likely degraded in the absence of its assembly partner (reviewed by Merkwirth C & Langer T, BBA, 2009) .

Thank you for your suggestion! Following your comment and the suggestions by reviewer #3, we have now performed a comprehensive analysis of the levels of Phb1 and Phb2 during development of peripheral nerves and of Phb2 levels in Phb1-SCKO animals. These new data are reported in our new Figure 2 and show that indeed, PHB2 protein is partially downregulated by deletion of Phb1. We adjusted the text throughout to account for the fact that the changes we see could be attributed to a reduction in both prohibitins.

4. Suppl. Fig 4: which nerve was examined here?

Sciatic nerves. This information has been added to the figure legend.

5. Suppl. Fig 6: phospho-specific antibodies are notoriously problematic with regard to specificity, and this also applies to anti-pHiston 3 (which the authors used to quantify mitotic activity). Here, staining for a proliferation-associated antigen such as Ki67 is perhaps much more informative (and technically much more robust).

Thank you for your suggestion! In our hands the p-H3 antibody has been very reliable and specific. But we now also report complete data using Ki67 in this figure and in the text. The data with Ki67 is similar to our previous data with p-H3.

6. Suppl. Fig 8: not clear, which timepoint (P20?, P40?) is shown.

P40. This information has been added to the figure legend.

7. There are many typos, just e.g. line 104 : ablation of Phb1 does not impairs, line 176: phagocytose; line 348: ISRIB; Figure 6G: mitochondrial dysfunction; Suppl Fig 7 and 9 (legend): Gaussian, and so on and so on (see my specific comment 2).

Thank you! This and other typos have been corrected. The new manuscript has been extensively revised by the authors (including English native speakers) and an additional native speaker who is not a co-

author.

Reviewer #3 (Remarks to the Author):

The manuscript by Della-Flora Nunes et al. represents a strong addition to the understanding of peripheral myelin maintenance by describing the important function of mitochondrial prohibitin1 in Schwann cell metabolism. The authors performed a comprehensive investigation in vivo, combining a wide array of elegant techniques and analyses. This manuscript will likely be seen as a major contribution to the ever-improving understanding of peripheral nervous system myelination and demyelinating peripheral neuropathies.

Previously, the authors identified novel molecules involved in axo-glial interaction and peripheral myelination, including members of the prohibitin family, and described the role of prohibitin 2 in Schwann cell radial sorting and myelination in vivo.

Thank you very much for the nice feedback and for your careful analysis of our manuscript.

In the present study, Della-Flora Nunes et al. found that unlike prohibitin 2, prohibitin 1 plays little to no role in developmental myelination, but is critical for peripheral myelin maintenance. This is a very interesting finding, however, I am concerned that because of the lack of expression data in the manuscript, the authors cannot rule out the hypothesis that prohibitin 1 and prohibitin 2 expression patterns are distinct, and that prohibitin 1 might be predominantly expressed in myelinating Schwann cells, and not at developmental stages (whereas prohibitin 2 might be expressed earlier over the course of Schwann cell development). I think it is worth investigating the transcript or protein levels of both paralogs (Phb1 and Phb2) at a timepoint corresponding to developmental myelination in WT animals, and at P20, which corresponds to the beginning of the Phb1 peripheral neuropathy. This would bring the missing piece needed to fully support the conclusion of this study.

Thank you for your suggestion. Following your comment and the comments by Reviewer #2, we performed a time course analysis of the mRNA and protein levels of both Phb1 and Phb2, confirming in mice our previous data obtained for Phb2 on rats⁴. In addition, we investigated what happens to the level of Phb2 in Phb1-SCKO mice. Overall, our data show that the expression of Phb1 and Phb2 is mostly but not completely co-regulated. In particular, the data show that both Phb1 and Phb2 are expressed more abundantly during development, suggesting that the difference in the phenotype after their ablation is not due to a predominant expression of Phb2 before myelination and a predominant expression of Phb1 after myelination. This new data is reported in our new Figure 2.

Another hypothesis is that a compensatory mechanism results in an increase in prohibitin 2 transcripts in Phb1-SCKO mice at least prior to and at the onset of myelination. In section 2.4 of the results, because prohibitins are required for the Raf-MEK-ERK cascade, the authors tested the expression and phosphorylation of ERK1/2 in Phb1-SCKO mice. As reported in Supplementary Figures 5A and B, they found little change in total and/or phosphorylated-ERK. Again, this could be explained by an upregulation of prohibitin 2, but I understand that investigating potential genetic compensatory mechanisms is beyond the scope of this paper.

Thank you for the suggestion. According to our new data in Figure 2D and 2E, it does not seem that deletion of Phb1 leads to a compensatory increase of *Phb2* transcripts. On the contrary, it seems that expression of Phb1 and Phb2 is mostly co-regulated, with *Phb2* mRNA levels not changing, but PHB2 protein levels also decreasing in Phb1-SCKO mice. Therefore, the lack of effect on the ERK1/2 phosphorylation suggests that, although important, prohibitins are not required for the Raf-MEK-ERK signaling cascade.

The following is not a concern, but more out of curiosity. When analyzing the Remak bundles, the authors noted the presence of Remak SCs retracting their processes and they hypothesized that they might undergo transdifferentiation to promote nerve repair. Did the authors observe elevated c-Jun in these Remak SCs?

Thank you for the interesting idea! Unfortunately, both the Remak SC marker (p75^{NTR}) and JUN antibodies validated in our lab were produced in rabbits. Thus, we could not perform a co-staining experiment.

One minor detail I would add relates to the graphs. The quantifications and graphs are all rigorous and convincing. Adding n.s. (or equivalent mention) when the comparisons are not statistically significant would help the reader sort the important message out of the data, which is quite considerable.

Thank you for the suggestion! We have now added n.s. to the graphs in the paper.

We hope that you find that our answers and the new data makes our paper suitable for publication at Nature Communications.

Thank you for your continued consideration and best wishes,

Gustavo Della Flora Nunes and Laura Feltri, on behalf of all authors.

References

- 1 Zhang, J. *et al.* Measuring energy metabolism in cultured cells, including human pluripotent stem cells and differentiated cells. *Nature Protocols* **7**, 1068-1085, doi:10.1038/nprot.2012.048 (2012).
- 2 Pooya, S. *et al.* The tumour suppressor LKB1 regulates myelination through mitochondrial metabolism. *Nat Commun* **5**, 4993, doi:10.1038/ncomms5993 (2014).
- 3 Javadov, S., Chapa-Dubocq, X. & Makarov, V. Different approaches to modeling analysis of mitochondrial swelling. *Mitochondrion* **38**, 58-70, doi:10.1016/j.mito.2017.08.004 (2018).
- 4 Poitelon, Y. *et al.* Spatial mapping of juxtacrine axo-glial interactions identifies novel molecules in peripheral myelination. *Nat Commun* **6**, 8303, doi:10.1038/ncomms9303 (2015).
- 5 Viader, A. *et al.* Schwann cell mitochondrial metabolism supports long-term axonal survival and peripheral nerve function. *J Neurosci* **31**, 10128-10140, doi:10.1523/jneurosci.0884-11.2011 (2011).
- 6 Cassidy-Stone, A. *et al.* Chemical Inhibition of the Mitochondrial Division Dynamin Reveals Its Role in Bax/Bak-Dependent Mitochondrial Outer Membrane Permeabilization. *Developmental Cell* **14**, 193-204, doi:10.1016/j.devcel.2007.11.019 (2008).
- 7 Huang, S. *et al.* Drp1-Mediated Mitochondrial Abnormalities Link to Synaptic Injury in Diabetes Model. *Diabetes* **64**, 1728-1742, doi:10.2337/db14-0758 (2015).

- 8 Brooks, C., Wei, Q., Cho, S.-G. & Dong, Z. Regulation of mitochondrial dynamics in acute kidney injury in cell culture and rodent models. *Journal of Clinical Investigation* **119**, 1275-1285, doi:10.1172/jci37829 (2009).

REVIEWERS' COMMENTS

Reviewer #1 (Remarks to the Author):

The authors have addressed my concerns.

Reviewer #2 (Remarks to the Author):

Overall, I appreciate the authors' efforts in addressing my (and the other referees') concerns. I realize that establishing a genetic model of Phb1 re-expression specifically in SCs (to test for Phb1 KO phenotype rescue) is somewhat beyond of what can be done in a reasonable time frame, and appreciate the alternative of using mDivi-1.

I am in favor of publication of the revised manuscript.

Reviewer #3 (Remarks to the Author):

I would like to thank the authors for taking my suggestions at heart and for integrating new supporting additional results into the revised manuscript. The careful time course analysis of both mRNA and protein levels of Phb1 and Phb2 really helps attribute the peripheral myelin maintenance phenotype to prohibitin 1 and addresses my concern that was shared with reviewer #2.

My other main concern was a hypothetical compensatory mechanism involving prohibitin 2 in Phb1-SCKO mice not assessed in the manuscript, which the authors have now convincingly ruled out. This study now brings a comprehensive and thorough analysis on the role of prohibitins in Schwann cell myelination and peripheral myelin maintenance.

Laura Fontenas

REVIEWERS' COMMENTS

Reviewer #1 (Remarks to the Author):

The authors have addressed my concerns.

Thank you very much for your careful revision of our paper!

Reviewer #2 (Remarks to the Author):

Overall, I appreciate the authors' efforts in addressing my (and the other referees') concerns. I realize that establishing a genetic model of Phb1 re-expression specifically in SCs (to test for Phb1 KO phenotype rescue) is somewhat beyond of what can be done in a reasonable time frame, and appreciate the alternative of using mDivi-1.

I am in favor of publication of the revised manuscript.

Thank you very much for your insights in suggestions on our manuscript!

Reviewer #3 (Remarks to the Author):

I would like to thank the authors for taking my suggestions at heart and for integrating new supporting additional results into the revised manuscript. The careful time course analysis of

both mRNA and protein levels of Phb1 and Phb2 really helps attribute the peripheral myelin maintenance phenotype to prohibitin 1 and addresses my concern that was shared with reviewer #2.

My other main concern was a hypothetical compensatory mechanism involving prohibitin 2 in Phb1-SCKO mice not assessed in the manuscript, which the authors have now convincingly ruled out. This study now brings a comprehensive and thorough analysis on the role of prohibitins in Schwann cell myelination and peripheral myelin maintenance.

Laura Fontenas

Thank you very much for your comments and suggestions, which we have now all incorporated and contributed to increase the quality of our manuscript.

Yours sincerely,

M.Laura Feltri M.D.

SUNY Distinguished Professor, Department of Biochemistry and Neuroscience

Acting Director, Hunter James Kelly Research Institute

State University of New York at Buffalo, School of Medicine and Biomedical Sciences

mlfeltri@buffalo.edu Off: 716-881-8969, Fax: 716-849-6651